# The Onset of Variance-Limited Behavior for Networks in the Lazy and Rich Regimes

**Alexander Atanasov**[* §‡] , **Blake Bordelon**[* †‡] , **Sabarish Sainathan** [†‡] **& Cengiz Pehlevan** [†‡]
§Department of Physics
†John A. Paulson School of Engineering and Applied Sciences
‡Center for Brain Science
 Harvard University
 Cambridge, MA 02138, USA
`{atanasov,blake_bordelon,cpehlevan}@g.harvard.edu`

## Abstract

For small training set sizes $P$, the generalization error of wide neural networks is well-approximated by the error of an infinite width neural network (NN), either in the kernel or mean-field/feature-learning regime. However, after a critical sample size $P^*$, we empirically find the finite-width network generalization becomes worse than that of the infinite width network. In this work, we empirically study the transition from infinite-width behavior to this *variance-limited* regime as a function of sample size $P$ and network width $N$. We find that finite-size effects can become relevant for very small dataset sizes on the order of $P^* \sim \sqrt{N}$ for polynomial regression with ReLU networks. We discuss the source of these effects using an argument based on the variance of the NN's final neural tangent kernel (NTK). This transition can be pushed to larger $P$ by enhancing feature learning or by ensemble averaging the networks. We find that the learning curve for regression with the final NTK is an accurate approximation of the NN learning curve. Using this, we provide a toy model which also exhibits $P^* \sim \sqrt{N}$ scaling and has $P$-dependent benefits from feature learning.

## 1 Introduction

Deep learning systems are achieving state of the art performance on a variety of tasks (Tan & Le, 2019; Hoffmann et al., 2022). Exactly how their generalization is controlled by network architecture, training procedure, and task structure is still not fully understood. One promising direction for deep learning theory in recent years is the infinite-width limit. Under a certain parameterization, infinite-width networks yield a kernel method known as the neural tangent kernel (NTK) (Jacot et al., 2018; Lee et al., 2019). Kernel methods are easier to analyze, allowing for accurate prediction of the generalization performance of wide networks in this regime (Bordelon et al., 2020; Canatar et al., 2021; Bahri et al., 2021; Simon et al., 2021). Infinite-width networks can also operate in the *mean-field* regime if network outputs are rescaled by a small parameter $\alpha$ that enhances feature learning (Mei et al., 2018; Chizat et al., 2019; Geiger et al., 2020b; Yang & Hu, 2020; Bordelon & Pehlevan, 2022).

While infinite-width networks provide useful limiting cases for deep learning theory, real networks have finite width. Analysis at finite width is more difficult, since predictions are dependent on the initialization of parameters. While several works have attempted to analyze feature evolution and kernel statistics at large but finite width (Dyer & Gur-Ari, 2020; Roberts et al., 2021), the implications of finite width on generalization are not entirely clear. Specifically, it is unknown at what value of the training set size $P$ the effects of finite width become relevant, what impact this critical $P$ has on the learning curve, and how it is affected by feature learning.

To identify the effects of finite width and feature learning on the deviation from infinite width learning curves, we empirically study neural networks trained across a wide range of output scales $\alpha$, widths $N$, and training set sizes $P$ on the simple task of polynomial regression with a ReLU neural network. Concretely, our experiments show the following:

---

*These authors contributed equally.

- Learning curves for polynomial regression transition exhibit significant finite-width effects very early, around $P \sim \sqrt{N}$. Finite-width NNs at large $\alpha$ are always outperformed by their infinite-width counterparts. We show this gap is driven primarily by variance of the predictor over initializations (Geiger et al., 2020a). Following prior work (Bahri et al., 2021), we refer to this as the *variance-limited regime*. We compare three distinct ensembling methods to reduce error in this regime.

- Feature-learning NNs show improved generalization both before and after the transition to the variance limited regime. Feature learning can be enhanced through re-scaling the output of the network by a small scalar $\alpha$ or by training on a more complex task (a higher-degree polynomial). We show that alignment between the final NTK and the target function on test data improves with feature learning and sample size.

- We demonstrate that the learning curve for the NN is well-captured by the learning curve for kernel regression with the *final* empirical NTK, eNTK$_f$ , as has been observed in other works (Vyas et al., 2022; Geiger et al., 2020b; Atanasov et al., 2021; Wei et al., 2022).

- Using this correspondence between the NN and the final NTK, we provide a cursory account of how fluctuations in the final NTK over random initializations are suppressed at large width $N$ and large feature learning strength. In a toy model, we reproduce several scaling phenomena, including the $P \sim \sqrt{N}$ transition and the improvements due to feature learning through an alignment effect.

We validate that these effects qualitatively persist in the realistic setting of wide ResNets Zagoruyko & Komodakis (2017) trained on CIFAR in appendix E.

Overall, our results indicate that the onset of finite-width corrections to generalization in neural networks become relevant when the scale of the variance of kernel fluctuations becomes comparable to the bias component of the generalization error in the bias-variance decomposition. The variance contribution to generalization error can be reduced both through ensemble averaging and through feature learning, which we show promotes higher alignment between the final kernel and the task. We construct a model of noisy random features which reproduces the essential aspects of our observations.

## 1.1 RELATED WORKS

Geiger et al. (2020a) analyzed the scaling of network generalization with the number of model parameters. Since the NTK fluctuates with variance $O(N^{-1})$ for a width $N$ network (Dyer & Gur-Ari, 2020; Roberts et al., 2021), they find that finite width networks in the lazy regime generically perform worse than their infinite width counterparts.

The scaling laws of networks over varying $N$ and $P$ were also studied, both empirically and theoretically by Bahri et al. (2021). They consider two types of learning curve scalings. First, they describe *resolution-limited* scaling, where either training set size or width are effectively infinite and the scaling behavior of generalization error with the other quantity is studied. There, the scaling laws can been obtained by the theory in Bordelon et al. (2020). Second, they analyze *variance-limited* scaling where width or training set size are fixed to a finite value and the other parameter is taken to infinity. While that work showed for any fixed $P$ that the learning curve converges to the infinite width curve as $O(N^{-1})$, these asymptotics do not predict, for fixed $N$, at which value of $P$ the NN learning curve begins to deviate from the infinite width theory. This is the focus of our work.

The contrast between rich and lazy networks has been empirically studied in several prior works. Depending on the structure of the task, the lazy regime can have either worse (Fort et al., 2020) or better (Ortiz-Jiménez et al., 2021; Geiger et al., 2020b) performance than the feature learning regime. For our setting, where the signal depends on only a small number of relevant input directions, we expect representation learning to be useful, as discussed in (Ghorbani et al., 2020; Paccolat et al., 2021b). Consequently, we posit and verify that the rich network will outperform the lazy one.

Our toy model is inspired by the literature on random feature models. Analysis of generalization for two layer networks at initialization in the limit of high dimensional data have been carried out using techniques from random matrix theory (Mei & Montanari, 2022; Hu & Lu, 2020; Adlam & Pennington, 2020a; Dhifallah & Lu, 2020; Adlam & Pennington, 2020b) and statistical mechanics (Gerace et al., 2020; d'Ascoli et al., 2020; d'Ascoli et al., 2020). Several of these works have identified that when $N$ is comparable to $P$, the network generalization error has a contribution

from variance over initial parameters. Further, they provide a theoretical explanation of the benefit of ensembling predictions of many networks trained with different initial parameters. Recently, Ba et al. (2022) studied regression with the hidden features of a two layer network after taking one step of gradient descent, finding significant improvements to the learning curve due to feature learning. Zavatone-Veth et al. (2022) analyzed linear regression for Bayesian deep linear networks with width $N$ comparable to sample size $P$ and demonstrated the advantage of training multiple layers compared to only training the only last layer, finding that feature learning advantage has leading correction of scale $(P/N)^2$ at small $P/N$.

## 2 PROBLEM SETUP AND NOTATION

We consider a supervised task with a dataset $\mathcal{D} = \{\boldsymbol{x}^\mu, y^\mu\}_{\mu=1}^P$ of size $P$. The pairs of data points are drawn from a population distribution $p(\boldsymbol{x}, y)$. Our experiments will focus on training networks to interpolate degree $k$ polynomials on the sphere (full details in Appendix A). For this task, the infinite width network learning curves can be found analytically. In particular at large $P$ the generalization error scales as $1/P^2$ (Bordelon et al., 2020). We take a single output feed-forward NN $\tilde{f}_\theta : \mathbb{R}^D \to \mathbb{R}$ with hidden width $N$ for each layer. We let $\theta$ denote all trainable parameters of the network. Using NTK parameterization (Jacot et al., 2018), the activations for an input $\boldsymbol{x}$ are given by

$$h_i^{(\ell)} = \frac{\sigma}{\sqrt{N}} \sum_{j=1}^N W_{ij}^{(\ell)} \varphi(h_j^{(\ell-1)}), \quad \ell = 2, \dots L, \quad h_i^{(1)} = \frac{\sigma}{\sqrt{D}} \sum_{j=1}^D W_{ij}^{(1)} x_j. \tag{1}$$

Here, the output of the network is $\tilde{f}_\theta = h_1^{(L)}$. We take $\varphi$ to be a positively homogenous function, in our case a ReLU nonlinearity, but this is not strictly necessary (Appendix C.2). At initialization we have $W_{ij} \sim \mathcal{N}(0, 1)$. Consequently, the scale of the output at initialization is $O(\sigma^L)$. As a consequence of the positive homogeneity of the network, the scale of the output is given by $\alpha = \sigma^L$. $\alpha$ controls the feature learning strength of a given NN. Large $\alpha$ corresponds to a *lazy* network while small $\alpha$ yields a *rich* network with feature movement. More details on how $\alpha$ controls feature learning are given in Appendix C.1 and C.2.

In what follows, we will denote the infinite width NTK limit of this network by NTK$_\infty$. We will denote its finite width linearization by $\text{eNTK}_0(\boldsymbol{x}, \boldsymbol{x}') := \sum_\theta \partial_\theta f(\boldsymbol{x}) \partial_\theta f(\boldsymbol{x}')|_{\theta=\theta_0}$, and we will denote its linearization around its final parameters $\theta_f$ by $\text{eNTK}_f(\boldsymbol{x}, \boldsymbol{x}') := \sum_\theta \partial_\theta f(\boldsymbol{x}) \partial_\theta f(\boldsymbol{x}')|_{\theta=\theta_f}$. Following other authors (Chizat et al., 2019; Adlam & Pennington, 2020a), we will take the output to be $f_\theta(\boldsymbol{x}) := \tilde{f}_\theta(\boldsymbol{x}) - \tilde{f}_{\theta_0}(\boldsymbol{x})$. Thus, at initialization the function output is 0. We explain this choice further in Appendix A. The parameters are then trained with full-batch gradient descent on a mean squared error loss. We denote the final network function starting from initialization $\theta_0$ on a dataset $\mathcal{D}$ by $f_{\theta_0, \mathcal{D}}^*(\boldsymbol{x})$ or $f^*$ for short. The generalization error is calculated using a held-out test set and approximates the population risk $E_g(f) := \langle (f(\boldsymbol{x}) - y)^2 \rangle_{\boldsymbol{x}, y \sim p(\boldsymbol{x}, y)}$.

## 3 EMPIRICAL RESULTS

In this section, we will study learning curves for ReLU NNs trained on polynomial regression tasks of varying degrees. We take our task to be learning $y = Q_k(\boldsymbol{\beta} \cdot \boldsymbol{x})$ where $\boldsymbol{\beta}$ is random vector of norm $1/D$ and $Q_k$ is the $k$th gegenbauer polynomial. We will establish the following key observations, which we will set out to theoretically explain in Section 4.

1. Both eNTK$_0$ and sufficiently lazy networks perform strictly worse than NTK$_\infty$, but the ensembled predictors approach the NTK$_\infty$ test error.

2. NNs in the feature learning regime of small $\alpha$ can outperform NTK$_\infty$ for an intermediate range of $P$. Over this range, the effect of ensembling is less notable.

3. Even richly trained finite width NNs eventually perform worse than NTK$_\infty$ at sufficiently large $P$. However, these small $\alpha$ feature-learning networks become variance-limited at larger $P$ than lazy networks. Once in the variance-limited regime, all networks benefit from ensembling over initializations.

4. For all networks, the transition to the variance-limited regime begins at a $P^*$ that scales sub-linearly with $N$. For polynomial regression, we find $P^* \sim \sqrt{N}$.

These findings support our hypothesis that finite width introduces variance in eNTK$_0$ over initializations, which ultimately leads to variance in the learned predictor and higher generalization error. Although we primarily focus on polynomial interpolation tasks in this paper, in Appendix F we provide results for wide ResNets trained on CIFAR and observe that rich networks also outperform lazy ones, and that lazy ones benefit more significantly from ensembling.

## 3.1 FINITE WIDTH EFFECTS CAUSE THE ONSET OF A VARIANCE LIMITED REGIME

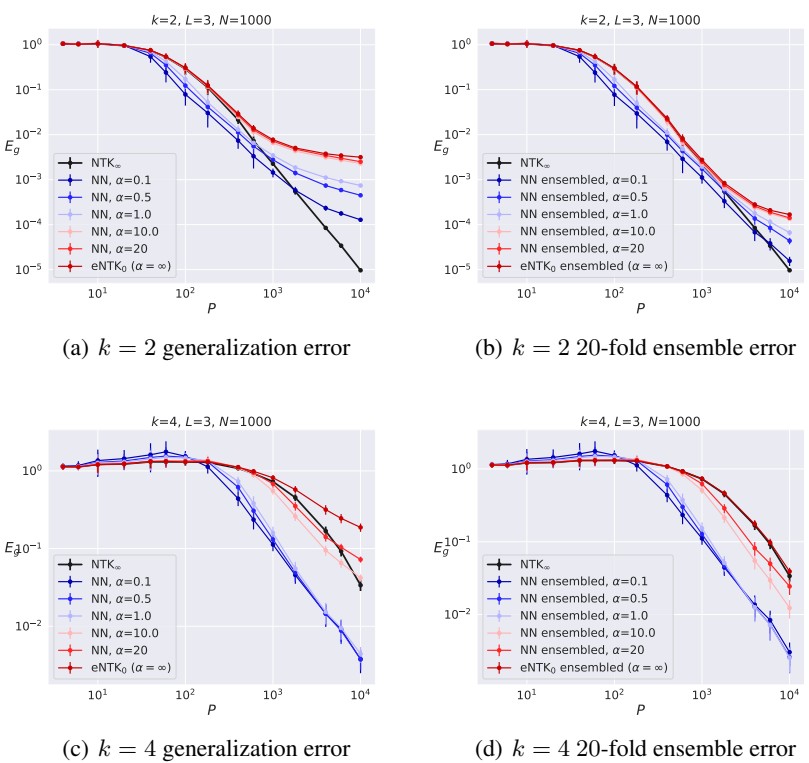

(a) $k = 2$ generalization error

(b) $k = 2$ 20-fold ensemble error

(c) $k = 4$ generalization error

(d) $k = 4$ 20-fold ensemble error

Figure 1: Generalization errors of depth $L = 3$ neural networks across a range of $\alpha$ values compared to NTK$_\infty$. The regression for NTK$_\infty$ was calculated using the Neural Tangents package (Novak et al., 2020). The exact scaling of NTK$_\infty$ is known to go asymptotically as $P^{-2}$ for this task. a) Lazy networks perform strictly worse than NTK$_\infty$ while rich networks can outperform it for an intermediate range of $P$ before their performance is also limited. b) Ensembling 20 networks substantially improves lazy network and eNTK$_0$ generalization, as well as asymptotic rich network generalization. This indicates that at sufficiently large $P$, these neural networks become limited by variance due to initialization. The error bars in a) and c) denote the variance due to both both training set and initialization. The error bars in b), d) denote the variance due to the train set.

In this section, we first investigate how finite width NN learning curves differ from infinite width NTK regression. In Figure 1 we show the generalization error $E_g\big(f^*_{\theta_0,\mathcal{D}}\big)$ for a depth 3 network with width $N = 1000$ trained on a quadratic $k = 2$ and quartic $k = 4$ polynomial regression task. Additional plots for other degree polynomials are provided in Appendix F. We sweep over $P$ to show the effect of more data on generalization, which is the main relationship we are interested in studying. For each training set size we sweep over a grid of 20 random draws of the train set and 20 random network initializations. This for 400 trained networks in total at each choice of $P, k, N, \alpha$. We see that a discrepancy arises at large enough $P$ where the neural networks begin to perform worse than NTK$_\infty$.

We probe the source of the discrepancy between finite width NNs and NTK$_\infty$ by ensemble averaging network predictions $\bar{f}_{\mathcal{D}}(\boldsymbol{x}) := \langle f^*_{\theta_0,\mathcal{D}}(\boldsymbol{x})\rangle_{\theta_0}$ over $E = 20$ initializations $\theta_0$. In Figures 1b and 1d, we calculate the error of $\bar{f}_{\mathcal{D}}(\boldsymbol{x})$, each trained on the same dataset. We then plot $E_g(\bar{f}_{\mathcal{D}})$. This

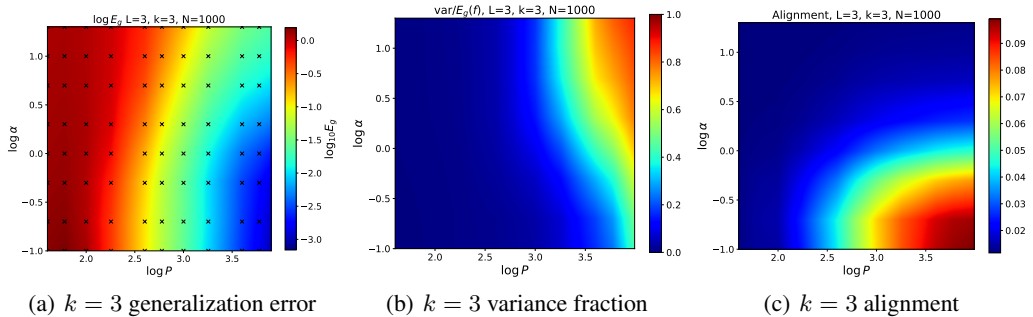

(a) $k = 3$ generalization error     (b) $k = 3$ variance fraction     (c) $k = 3$ alignment

Figure 2: Phase plots in the $P, \alpha$ plane of a) The log generalization error $\log_{10} E_g(f^\star)$, b) The fraction of generalization error removed by ensembling $1 - E_g(\bar{f}^\star)/E_g(f^\star)$, c) Kernel-task alignment measured by $\frac{\boldsymbol{y}^T K_f \boldsymbol{y}}{|\boldsymbol{y}|^2 \mathrm{Tr} K_f}$ where $\boldsymbol{y}$ and $K_f$ are evaluated on test data. We have plotted 'x' markers in a) to show the points where the NNs were trained.

ensembled error approximates the bias in a bias-variance decomposition (Appendix B). Thus, any gap between 1 (a) and 1 (b) is driven by variance of $f_{\theta, \mathcal{D}}$ over $\theta$.

We sharpen these observations with phase plots of NN generalization, variance and kernel alignment over $P, \alpha$, as shown in Figure 2. In Figure 2a, generalization for NNs in the rich regime (small $\alpha$) have lower final $E_g$ than lazy networks. As the dataset grows, the fraction of $E_g$ due to initialization variance (that is, the fraction removed by ensembling) strictly increases (2 (b)). We will show why this effect occurs in section 3.2. Figure 2b shows that, at any fixed $P$, the variance is lower for small $\alpha$. To measure the impact of feature learning on the eNTK$_f$, we plot its alignment with the target function, measured as $\frac{\boldsymbol{y}^\top \boldsymbol{K} \boldsymbol{y}}{|\boldsymbol{y}|^2 \mathrm{Tr} \boldsymbol{K}}$ for a test set of targets $[\boldsymbol{y}]_\mu$ and kernel $[\boldsymbol{K}]_{\mu\nu} = \mathrm{eNTK}_f(\boldsymbol{x}_\mu, \boldsymbol{x}_\nu)$. Alignment of the kernel with the target function is known to be related to good generalization (Canatar et al., 2021). In Section 4, we revisit these effects in a simple model which relates kernel alignment and variance reduction.

In addition to initialization variance, variance over dataset $\mathcal{D}$ contributes to the total generalization error. Following (Adlam & Pennington, 2020b), we discuss a symmetric decomposition of the variance in Appendix B, showing the contribution from dataset variance and the effects of bagging. We find that most of the variance in our experiments is due to initialization.

We show several other plots of the results of these studies in the appendix. We show the effect of bagging (Figure 7), phase plots of different degree target functions (Figures 10, 9), phase plots over $N, \alpha$ (Figure 11) and a comparison of network predictions against the initial and final kernel regressors (Figures 18, 19).

## 3.2 FINAL NTK VARIANCE LEADS TO GENERALIZATION PLATEAU

In this section, we show how the variance over initialization can be interpreted as kernel variance in both the rich and lazy regimes. We also show how this implies a plateau for the generalization error.

To begin, we demonstrate empirically that all networks have the same generalization error as kernel regression solutions with their *final* eNTKs. At large $\alpha$, the initial and the final kernel are already close, so this follows from earlier results of Chizat et al. (2019). In the rich regime, the properties of the eNTK$_f$ have been studied in several prior works. Several have empirically demonstrated that the eNTK$_f$ is a good match to the final network predictor for a trained network (Long, 2021; Vyas et al., 2022; Wei et al., 2022) while others have given conditions under which such an effect would hold true (Atanasov et al., 2021; Bordelon & Pehlevan, 2022). We comment on this in appendix C.4. We show in Figure 3 how the final network generalization error matches the generalization error of eNTK$_f$. As a consequence, we can use eNTK$_f$ to study the observed generalization behavior.

Next, we relate the variance of the final predictor $f^*_{\theta_0, \mathcal{D}}$ to the corresponding infinite width network $f^\infty_{\mathcal{D}}$. The finite size fluctuations of the kernel at initialization have been studied in (Dyer & Gur-Ari, 2020; Hanin & Nica, 2019; Roberts et al., 2021). The variance of the kernel elements has been

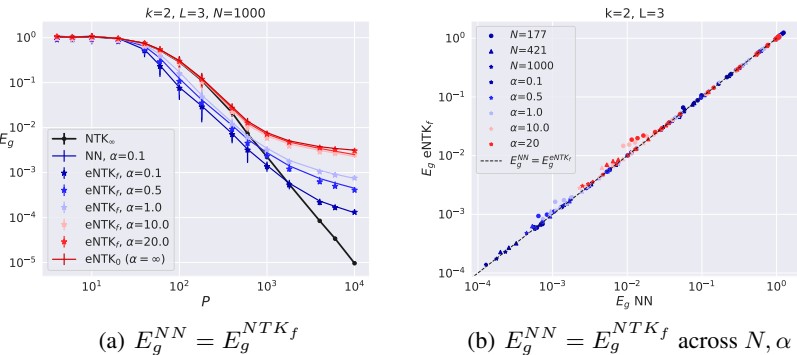

(a) $E_g^{NN} = E_g^{NTK_f}$  (b) $E_g^{NN} = E_g^{NTK_f}$ across $N, \alpha$

Figure 3: Kernel regression with eNTK$_f$ reproduces the learning curves of the NN with high fidelity. (a) Learning curves across different laziness settings $\alpha$ in a width 1000 network. The solid black curve is the infinite width network. Colored curves are the NN generalizations. Stars represent the eNTK$_f$s, and lie on top of the corresponding NN learning curves. (b) The agreement of generalizations between NNs and eNTK$_f$s across different $N$ and $\alpha$. Here the colors denote different $\alpha$ values while the dot, triangle and star markers denote networks of $N = \{177, 421, 1000\}$ respectively.

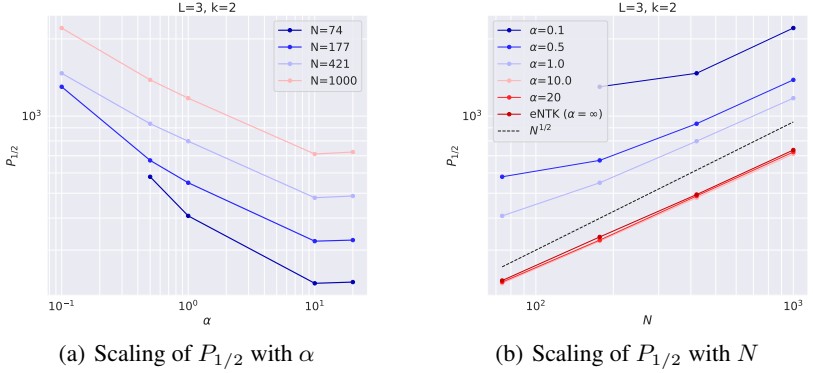

(a) Scaling of $P_{1/2}$ with $\alpha$  (b) Scaling of $P_{1/2}$ with $N$

Figure 4: Critical sample size $P_{1/2}$ measures the onset of the variance limited regime as a function of $\alpha$ at fixed $N$. (a) More feature learning (small $\alpha$) delays the transition to the variance limited regime. (b) $P_{1/2}$ as a function of $N$ for fixed $\alpha$ has roughly $P_{1/2} \sim \sqrt{N}$ scaling.

shown to scale as $1/N$. We perform the following bias-variance decomposition: Take $f_{\theta_0,\mathcal{D}}$ to be the eNTK$_0$ predictor, or a sufficiently lazy network trained to interpolation on a dataset $\mathcal{D}$. Then,

$$\langle (f_{\theta_0,\mathcal{D}}^*(\boldsymbol{x}) - y)^2 \rangle_{\theta_0,\mathcal{D},\boldsymbol{x},y} = \langle (f_{\mathcal{D}}^\infty(\boldsymbol{x}) - y)^2 \rangle_{\mathcal{D},\boldsymbol{x},y} + O(1/N). \tag{2}$$

We demonstrate this equality using a relationship between the infinite-width network and an infinite ensemble of finite-width networks derived in Appendix B. There we also show that the $O(1/N)$ term is strictly positive for sufficiently large $N$. Thus, for lazy networks of sufficiently large $N$, finite width effects lead to strictly worse generalization error. The decomposition in Equation 2 continues to hold for rich networks at small $\alpha$ if $f^\infty$ is interpreted as the infinite-width mean field limit. In this case one can show that ensembles of rich networks are approximating an infinite width limit in the mean-field regime. See Appendix B for details.

### 3.3 FEATURE LEARNING DELAYS VARIANCE LIMITED TRANSITION

We now consider how feature learning alters the onset of the variance limited regime, and how this onset scales with $\alpha, N$. We define the onset of the variance limited regime to take place at the value $P^* = P_{1/2}$ where over half of the generalization error is due to variance over initializations. Equivalently we have $E_g(\bar{f}^*)/E_g(f^*) = 1/2$. By using an interpolation method together with bisection, we solve for $P_{1/2}$ and plot it in Figure 4.

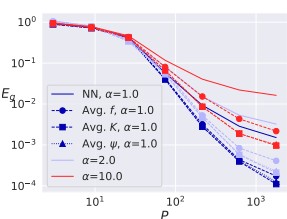
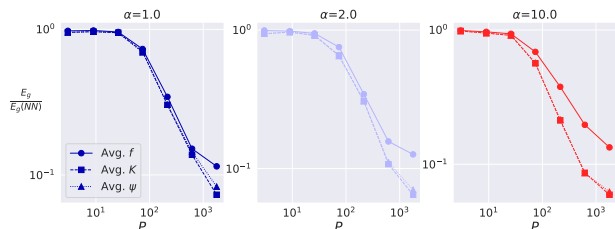

(a) Ensembling Methods

(b) Reduction in $E_g$ for Each Ensembling Technique

Figure 5: The random feature model suggests three possible types of ensembling: averaging the output function $f(\boldsymbol{x}, \theta)$, averaging eNTK$_f$ $K(\boldsymbol{x}, \boldsymbol{x}'; \theta)$, and averaging the induced features $\boldsymbol{\psi}(\boldsymbol{x}, \theta)$. We analyze these ensembling methods for a $k = 1$ task with a width $N = 100$ ReLU network. (a) While all ensembling methods improve generalization, averaging either the kernel $\langle K \rangle$ or features $\langle \psi \rangle$ gives a better improvement to generalization than averaging the output function $\langle f \rangle$. Computing final kernels for many richly trained networks and performing regression with this averaged kernel gives the best performance. (b) We plot the relative error of each ensembling method against the single init neural network. The gap between ensembling and the single init NN becomes evident for sufficiently large $P \sim P_{1/2}$. For small $\alpha$, all ensembling methods perform comparably, while for large $\alpha$ ensembling the kernel or features gives much lower $E_g$ than averaging the predictors.

Figure 4b shows that $P_{1/2}$ scales as $\sqrt{N}$ for this task. In the next section, we shall show that this scaling is governed by the fact that $P_{1/2}$ is close to the value where the infinite width network generalization curve $E_g^\infty$ is equal to the variance of eNTK$_f$. In this case the quantities to compare are $E_g^\infty \approx P^{-2}$ and $\mathrm{Var}\, \mathrm{eNTK}_f \approx N^{-1}$.

We can understand the delay of the variance limited transition, as well as the lower value of the final plateau using a mechanistic picture similar to the effect observed in Atanasov et al. (2021). In that setting, under small initialization, the kernel follows a deterministic trajectory, picking up a low rank component in the direction of the train set targets $\boldsymbol{y}\boldsymbol{y}^\top$, and then changing only in scale as the network weights grow to interpolate the dataset. In their case, for initial output scale $\sigma^L$, eNTK$_f$ is deterministic up to a variance of $O(\sigma)$. In our case, the kernel variance at initialization scales as $\sigma^{2L}/N$. As $\sigma \to 0$ the kernel's trajectory becomes deterministic up to a variance term scaling with $\sigma$ as $O(\sigma)$, which implies that the final predictor also has a variance scaling as $O(\sigma)$.

## 4 SIGNAL PLUS NOISE CORRELATED FEATURE MODEL

In Section 3.2 we have shown that in both the rich and lazy regimes, the generalization error of the NN is well approximated by the generalization of a kernel regression solution with eNTK$_f$. This finding motivates an analysis of the generalization of kernel machines which depend on network initialization $\theta_0$. Unlike many analyses of random feature models which specialize to two layer networks and focus on high dimensional Gaussian random data (Mei & Montanari, 2022; Adlam & Pennington, 2020a; Gerace et al., 2020; Ba et al., 2022), we propose to analyze regression with the eNTK$_f$ for more general feature structures. This work builds on the kernel generalization theory for kernels developed with statistical mechanics (Bordelon et al., 2020; Canatar et al., 2021; Simon et al., 2021; Loureiro et al., 2021). We will attempt to derive approximate learning curves in terms of the eNTK$_f$'s signal and noise components, which provide some phenomenological explanations of the onset of the variance limited regime and the benefits of feature learning. Starting with the final NTK $K_{\theta_0}(\boldsymbol{x}, \boldsymbol{x}')$ which depends on the random initial parameters $\theta_0$, we project its square root $K_{\theta_0}^{1/2}(\boldsymbol{x}, \boldsymbol{x}')$ (as defined in equation 32) on a fixed basis $\{b_k(x)\}_{k=1}^\infty$ orthonormal with respect to $p(\boldsymbol{x})$. This defines a feature map

$$\psi_k(\boldsymbol{x}, \theta_0) = \int d\boldsymbol{x}' p(\boldsymbol{x}') K_{\theta_0}^{1/2}(\boldsymbol{x}, \boldsymbol{x}') b_k(\boldsymbol{x}') \ , \ k \in \{1, ..., \infty\}. \tag{3}$$

The kernel can be reconstructed from these features $K_{\theta_0}(\boldsymbol{x}, \boldsymbol{x}') = \sum_k \psi_k(\boldsymbol{x}, \theta_0)\psi_k(\boldsymbol{x}', \theta_0)$. The kernel interpolation problem can be solved by performing linear regression with features $\boldsymbol{\psi}(\boldsymbol{x}, \theta_0)$. Here, $\boldsymbol{w}(\theta_0) = \lim_{\lambda \to 0} \mathrm{argmin}_{\boldsymbol{w}} \sum_{\mu=1}^P [\boldsymbol{w} \cdot \boldsymbol{\psi}(\boldsymbol{x}_\mu, \theta_0) - y_\mu]^2 + \lambda |\boldsymbol{w}|^2$. The learned function

$f(\boldsymbol{x}, \theta_0) = \boldsymbol{w}(\theta_0) \cdot \boldsymbol{\psi}(\boldsymbol{x}, \theta_0)$ is the minimum norm interpolator for the kernel $K(\boldsymbol{x}, \boldsymbol{x}'; \theta_0)$ and matches the neural network learning curve as seen in Section 3.2. In general, since the rank of $K$ is finite for a finite size network, the $\psi_k(\boldsymbol{x}, \theta_0)$ have correlation matrix of finite rank $N_{\mathcal{H}}$. Since the target function $y$ does not depend on the initialization $\theta_0$, we decompose it in terms of a fixed set of features $\boldsymbol{\psi}_M(\boldsymbol{x}) \in \mathbb{R}^M$ (for example, the first $M$ basis functions $\{b_k\}_{k=1}^M$). In this random feature model, one can interpret the initialization-dependent fluctuations in $K(\boldsymbol{x}, \boldsymbol{x}'; \theta_0)$ as generating fluctuations in the features $\boldsymbol{\psi}(\boldsymbol{x}, \theta_0)$ which induce fluctuations in the learned network predictor $f(\boldsymbol{x}, \theta_0)$. To illustrate the relative improvements to generalization from denoising these three different objects, in Figure 5, we compare averaging the final kernel $K$, averaging the induced features $\psi$, and averaging network predictions $f$ directly. For all $\alpha$, all ensembling methods provide improvements over training a single NN. However, we find that averaging the kernel directly and performing regression with this kernel exhibits the largest reduction in generalization error. Averaging features performs comparably. However, ensemble averaging network predictors does not perform as well as either of these other two methods. The gap between ensembling methods is more significant in the lazy regime (large $\alpha$) and is negligible in the rich regime (small $\alpha$).

## 4.1 TOY MODELS AND APPROXIMATE LEARNING CURVES

To gain insight into the role of feature noise, we characterize the test error associated with a Gaussian covariate model in a high dimensional limit $P, M, N_{\mathcal{H}} \to \infty$ with $\alpha = P/M, \eta = N_{\mathcal{H}}/M$.

$$y = \frac{1}{\sqrt{M}} \boldsymbol{\psi}_M \cdot \boldsymbol{w}^*, f = \frac{1}{\sqrt{M}} \boldsymbol{\psi} \cdot \boldsymbol{w}, \ \boldsymbol{\psi} = \boldsymbol{A}(\theta_0)\boldsymbol{\psi}_M + \boldsymbol{\epsilon}, \ \begin{bmatrix} \boldsymbol{\psi}_M \\ \boldsymbol{\epsilon} \end{bmatrix} \sim \mathcal{N}\left(0, \begin{bmatrix} \boldsymbol{\Sigma}_M & 0 \\ 0 & \boldsymbol{\Sigma}_\epsilon \end{bmatrix}\right) \quad (4)$$

This model was also studied by Loureiro et al. (2021) and subsumes the classic two layer random feature models of prior works (Hu & Lu, 2020; Adlam & Pennington, 2020a; Mei & Montanari, 2022). The expected generalization error for any distribution of $\boldsymbol{A}(\theta_0)$ has the form

$$\mathbb{E}_{\theta_0} E_g(\theta_0) = \mathbb{E}_{\boldsymbol{A}} \frac{1}{1-\gamma} \frac{1}{M} \boldsymbol{w}^* \boldsymbol{\Sigma}_M^{1/2} \left[ \boldsymbol{I} - \hat{q}\boldsymbol{\Sigma}_s^{1/2} \boldsymbol{A}^\top \boldsymbol{G} \boldsymbol{A} \boldsymbol{\Sigma}_s^{1/2} - \hat{q}\boldsymbol{\Sigma}_s^{1/2} \boldsymbol{A}^\top \boldsymbol{G}^2 \boldsymbol{A} \boldsymbol{\Sigma}_s^{1/2} \right] \boldsymbol{\Sigma}_M^{1/2} \boldsymbol{w}^*$$

$$\boldsymbol{G} = \left( \boldsymbol{I} + \hat{q}\boldsymbol{A}\boldsymbol{\Sigma}_M \boldsymbol{A}^\top + \hat{q}\boldsymbol{\Sigma}_\epsilon \right)^{-1}, \ \hat{q} = \frac{\alpha}{\lambda + q}, \ q = \mathrm{Tr}\boldsymbol{G}[\boldsymbol{A}\boldsymbol{\Sigma}_M \boldsymbol{A}^\top + \boldsymbol{\Sigma}_\epsilon], \quad (5)$$

where $\alpha = P/M$ and $\gamma = \frac{\alpha}{(\lambda+q)^2}\mathrm{Tr}\boldsymbol{G}^2[\boldsymbol{A}\boldsymbol{\Sigma}_M \boldsymbol{A}^\top + \boldsymbol{\Sigma}_\epsilon]^2$. Details of the calculation can be found in Appendix D. We also provide experiments showing the predictive accuracy of the theory in Figure 6. In general, we do not know the induced distribution of $\boldsymbol{A}(\theta_0)$ over disorder $\theta_0$. In Appendix D.5, we compute explicit learning curves for a simple toy model where $\boldsymbol{A}(\theta_0)'s$ entries as i.i.d. Gaussian over the random initialization $\theta_0$. A similar random feature model was recently analyzed with diagrammatic techniques by Maloney et al. (2022). In the high dimensional limit $M, P, N_{\mathcal{H}} \to \infty$ with $P/M = \alpha, N_{\mathcal{H}}/M = \eta$, our replica calculation demonstrates that test error is self-averaging (the same for every random instance of $\boldsymbol{A}$) which we describe in Appendix D.5 and Figure 16.

## 4.2 EXPLAINING FEATURE LEARNING BENEFITS AND ERROR PLATEAUS

Using this theory, we can attempt to explain some of the observed phenomena associated with the onset of the variance limited regime. First, we note that the kernels exhibit fluctuations over initialization with variance $O(1/N)$, either in the lazy or rich regime. In Figure 6 (a), we show learning curves for networks of different widths in the lazy regime. Small width networks enter the variance limited regime earlier and have higher error. Similarly, if we alter the scale of the noise $\boldsymbol{\Sigma}_\epsilon = \sigma_\epsilon^2 \boldsymbol{A}\boldsymbol{\Sigma}_M \boldsymbol{A}^\top$ in our toy model, the corresponding transition time $P_{1/2}$ is smaller and the asymptotic error is higher. In Figure 6 (c), we show that our theory also predicts the onset of the variance limited regime at $P_{1/2} \sim \sqrt{N}$ if $\sigma_\epsilon^2 \sim N^{-1}$. We stress that this scaling is a consequence of the structure of the task. Since the target function is an eigenfunction of the kernel, the infinite width error goes as $1/P^2$ (Bordelon et al., 2020). Since variance scales as $1/N$, bias and variance become comparable at $P \sim \sqrt{N}$. Often, realistic tasks exhibit power law decays where $E_g^{N=\infty} = P^{-\beta}$ with $\beta < 2$ (Spigler et al., 2020; Bahri et al., 2021), where we'd expect a transition around $P_{1/2} \sim N^{1/\beta}$.

Using our model, we can also approximate the role of feature learning as enhancement in the signal correlation along task-relevant eigenfunctions. In Figure 6 (d) we plot the learning curves for networks trained with different levels of feature learning, controlled by $\alpha$. We see that feature learning leads to improvements in the learning curve both before and after onset of variance limits. In Figure

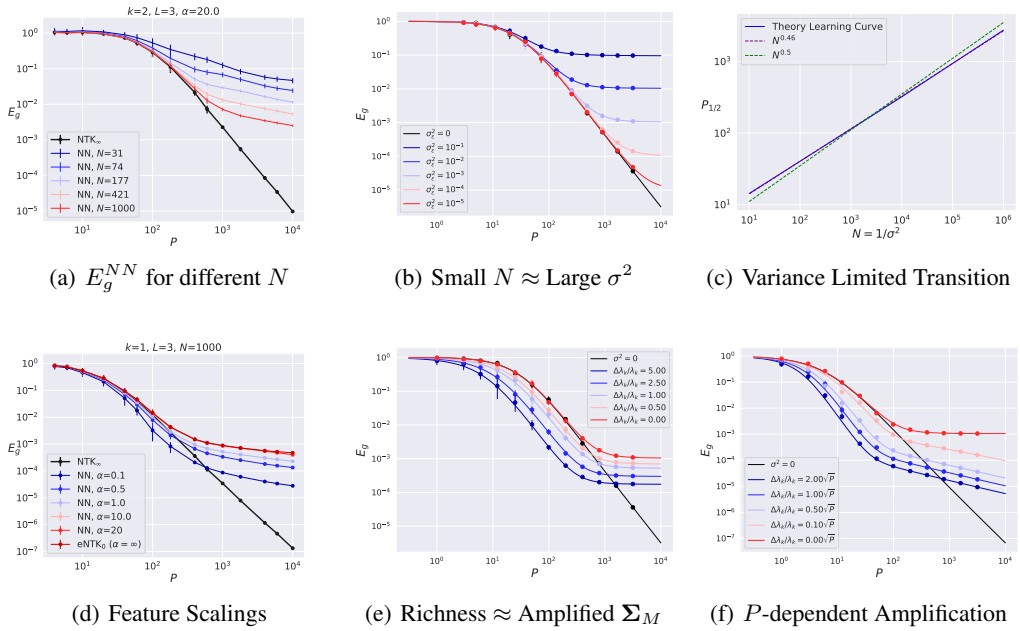

(a) $E_g^{NN}$ for different $N$      (b) Small $N \approx$ Large $\sigma^2$      (c) Variance Limited Transition

(d) Feature Scalings      (e) Richness $\approx$ Amplified $\boldsymbol{\Sigma}_M$      (f) $P$-dependent Amplification

Figure 6: A toy model of noisy features reproduces qualitative dependence of learning curves on kernel fluctuations and feature learning. (a) The empirical learning curves for networks of varying width $N$ at large $\alpha$. (b) Noisy kernel regression learning curve with noise $\boldsymbol{\Sigma}_\epsilon = \sigma_\epsilon^2 \boldsymbol{\Sigma}_M$ and $\boldsymbol{A}$ is a projection matrix preserving 20-k top eigenmodes of $\boldsymbol{\Sigma}_M$, which was computed from the NTK$_\infty$ for a depth 3 ReLU network. (c) This toy model reproduces the approximate scaling of the transition sample size $P_{1/2} \sim N^{1/2}$ if $\sigma_\epsilon^2 \sim N^{-1}$. (d) NNs trained with varying richness $\alpha$. Small $\alpha$ improves the early learning curve and asymptotic behavior. (e) Theory curves for a kernel with amplified eigenvalue $\lambda_k \to \lambda_k + \Delta\lambda_k$ for the target eigenfunction. This amplification mimics the effect of enhanced kernel alignment in the low $\alpha$ regime. Large amplification improves generalization performance. (f) $P$-dependent alignment where $\Delta\lambda_k \sim \sqrt{P}$ gives a better qualitative match to (d).

6 (e)-(f), we plot the theoretical generalization for kernels with enhanced signal eigenvalue for the task eigenfunction $y(\boldsymbol{x}) = \phi_k(\boldsymbol{x})$. This enhancement, based on the intuition of kernel alignment, leads to lower bias and lower asymptotic variance. However, this model does not capture the fact that feature learning advantages are small at small $P$ and that the slopes of the learning curves are different at different $\alpha$. Following the observation of Paccolat et al. (2021a) that kernel alignment can occur with scale $\sqrt{P}$, we plot the learning curves for signal enhancements that scale as $\sqrt{P}$. Though this toy model reproduces the onset of the variance limited regime $P_{1/2}$ and the reduction in variance due to feature learning, our current result is not the complete story. A more refined future theory could use the structure of neural architecture to constrain the structure of the $\boldsymbol{A}$ distribution.

## 5   CONCLUSION

We performed an extensive empirical study for deep ReLU NNs learning a fairly simple polynomial regression problems. For sufficiently large dataset size $P$, all neural networks under-perform the infinite width limit, and we demonstrated that this worse performance is driven by initialization variance. We show that the onset of the variance limited regime can occur early in the learning curve with $P_{1/2} \sim \sqrt{N}$, but this can be delayed by enhancing feature learning. Finally, we studied a simple random-feature model to attempt to explain these effects and qualitatively reproduce the observed behavior, as well as quantitatively reproducing the relevant scaling relationship for $P_{1/2}$. This work takes a step towards understanding scaling laws in regimes where finite-size networks undergo feature learning. This has implications for how the choice of initialization scale, neural architecture, and number networks in an ensemble can be tuned to achieve optimal performance under a fixed compute and data budget.

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

## A  DETAILS ON EXPERIMENTS

We generated the dataset $\mathcal{D} = \{\boldsymbol{x}^\mu, y^\mu\}_{\mu=1}^P$ by sampling $\boldsymbol{x}^\mu$ uniformly on $\mathbb{S}^{D-1}$, the unit sphere in $\mathbb{R}^D$. $\tilde{y}$ was then generated as a Gegenbauer polynomial of degree $k$ of a 1D projection of $\boldsymbol{x}$, $\tilde{y} = Q_k(\boldsymbol{\beta} \cdot \boldsymbol{x})$. Because the scale of the output of the neural network relative to the target is a central quantity in this work, it is especially important to make sure the target is appropriately scaled to unit norm. We did this by defining the target to be $y = \tilde{y}/\sqrt{\langle Q_k(\boldsymbol{\beta} \cdot \boldsymbol{x})^2\rangle_{\boldsymbol{x}\sim\mathbb{S}^{D-1}}}$. The denominator can be easily and accurately approximated by Monte Carlo sampling.

We used JAX (Bradbury et al., 2018) for all neural network training. We built multi-layer percep-trons (MLPs) of depth 2 and 3. Most of the results are reported for depth 3 perceptrons, where there is a separation between the width of the network $N$ and the number of parameters $N^2$. Sweeping over more depths and architectures is possible, but because of the extensive dimensionality of the hyperparameter search space, we have not yet experimented with deeper networks.

We considered MLPs with no bias terms. Since the Gegenbauer polynomials are mean zero, we do not need biases to fit the training set and generalize well. We have also verified that adding trainable biases does not change the final results in any substantial way.

As mentioned in the main text, we consider the final output function to be the initial network output minus the output at initialization:

$$f_\theta(\boldsymbol{x}) = \tilde{f}_\theta(\boldsymbol{x}) - \tilde{f}_{\theta_0}(\boldsymbol{x}). \tag{6}$$

Here, only $\theta$ is differentiated through, while $\theta_0$ is held fixed. The rationale for this choice is that without this subtraction, in the lazy limit the trained neural network output can be written as

$$\tilde{f}_\theta^*(\boldsymbol{x}) = \tilde{f}_{\theta_0}(\boldsymbol{x}) + \sum_{\mu\nu} \boldsymbol{k}_\mu(\boldsymbol{x})[\boldsymbol{K}^{-1}]_{\mu\nu}(y^\nu - \tilde{f}_{\theta_0}(\boldsymbol{x})). \tag{7}$$

This is the same as doing eNTK$_0$ regression on the shifted targets $y^\mu - \tilde{f}_{\theta_0}(\boldsymbol{x})$. At large initialization the shift $\tilde{f}_{\theta_0}(\boldsymbol{x})$ amounts to adding random, initialization-dependent noise to the targets. By instead performing the subtraction, the lazy limit can be interpreted as a kernel regression on the targets themselves, which is preferable.

We trained this network with full batch gradient descent with a learning rate $\eta$ so that

$$\Delta\theta = -\eta\nabla_\theta\mathcal{L}(\mathcal{D}, \theta),$$
$$\mathcal{L}(\mathcal{D}, \theta) := \frac{1}{P}\sum_{\mu=1}^P |f_\theta(\boldsymbol{x}^\mu) - y^\mu|^2. \tag{8}$$

Each network was trained to an interpolation threshold of $10^{-6}$. If a network could not reach this threshold in under 30k steps, we checked if the training error was less than 10 times the generalization error. If this was not satisfied, then that run of the network was discarded.

For each fixed $P, k$, we generated 20 independent datasets. For each fixed $N, \alpha$ we generated 20 independent neural network initializations. This $20 \times 20$ table yields a total of 400 neural networks trained on every combination of initialization and dataset choice.

The infinite width network predictions were calculated using the Neural Tangents package (Novak et al., 2020). The finite width eNTK$_0$s were also calculated using the empirical methods in Neural Tangents. They were trained to interpolation using the `gradient_descent_mse` method. This is substantially faster than training the linearized model using standard full-batch gradient descent, which we have found to take a very long time for most networks. We use the same strategy for the eNTK$_f$s.

For the experiments in the main text, we have taken the input dimension to be $D = 10$ and sweep over $k = 1, 2, 3, 4$. We swept over 15 values $P$ in logspace from size 30 to size 10k, and over 6 values of $N$ in logspace from size 30 to size 2150. We then swept over alpha values $0.1, 0.5, 1.0, 10.0, 20.0$. Depending on $\alpha, N$, we tuned the learning rate $\eta$ of the network small enough to stay close to the gradient flow limit, but allow for the interpolation threshold to be feasibly reached.

For each of the 1800 settings of $P, N, \alpha, k$ and each of the 400 networks, 400 eNTK$_0$s, 400 eNTK$_f$s, and 20 NTK$_\infty$s, the generalization error was saved, as well as a vector of $\hat{y}$ predictions on a test set

of 2000 points. In addition, for the neural networks we saved both initial and final parameters. All are saved as lists of numpy arrays in a directory of about 1TB. We plan to make the results of our experiments publicly accessible, alongside the code to generate them.

## A.1 CIFAR EXPERIMENTS

We apply the same methodology of centering the network and allowing $\alpha$ to control the degree of laziness by redefining

$$f_\theta(\boldsymbol{x}) = \alpha(\tilde{f}_\theta(\boldsymbol{x}) - \tilde{f}_{\theta_0}(\boldsymbol{x})). \tag{9}$$

We consider the task of binary classification for CIFAR-10. In order to allow $P$ to become large we divide the data into two classes: animate and inanimate objects. We choose to subsample eight classes and superclass them into two: (cat, deer, dog, horse) vs (airplane, automobile, ship, truck). Each superclass consists of 20,000 training examples and 4,000 test examples retrieved from the CIFAR-10 dataset.

On subsets of this dataset, we train wide residual networks (ResNets) Zagoruyko & Komodakis (2017) of width 64 and block size 1 with the NTK parameterization Jacot et al. (2018) on this task using mini-batch gradient descent with batch size of 256 and MSE loss. Step sizes are governed by the Adam optimizer Kingma & Ba (2014) with initial learning rate $\eta_0 = 10^{-3}$. Every network is trained for 24,000 steps, such that under nearly all settings of $\alpha$ and dataset size the network has attained infinitesimal train loss.

We sweep $\alpha$ from $10^{-3}$ to $10^0$ and $P$ from $2^9$ to $2^{15}$. For each value of $P$, we randomly sample five training datasets of size $P$ and compute ensembles of size 20. For each network in an ensemble the initialization and the order of the training data is randomly chosen independently of those for the other networks.

# B FINE-GRAINED BIAS-VARIANCE DECOMPOSITION

## B.1 FINE GRAINED DECOMPOSITION OF GENERALIZATION ERROR

Let $\mathcal{D}$ be a dataset of $(\boldsymbol{x}^\mu, y^\mu)_{\mu=1}^P \sim p(\boldsymbol{x}, y)$ viewed as a random variable. Let $\theta_0$ represent the initial parameters of a neural network, viewed as a random variable. In the case of no label noise, as in section 2.2.1 of Adlam & Pennington (2020b), we derive the symmetric decomposition of the generalization error in terms of the variance due to initialization and the variance due to the dataset. We have

$$E_g(f_{\theta_0,\mathcal{D}}^*) = \langle (f_{\theta_0,\mathcal{D}}^*(\boldsymbol{x}) - y)^2 \rangle_{\boldsymbol{x},y} = \langle (\langle f_{\theta_0,\mathcal{D}}^*(\boldsymbol{x}) \rangle_{\theta_0,\mathcal{D}} - y)^2 \rangle_{\boldsymbol{x},y} + \mathbb{E}_{\boldsymbol{x}} \mathrm{Var}_{\theta_0,\mathcal{D}} f_{\theta_0,\mathcal{D}}^*(y)$$
$$= \mathrm{Bias}^2 + V_\mathcal{D} + V_{\theta_0} + V_{\mathcal{D},\theta_0}. \tag{10}$$

Here we have defined

$$\mathrm{Bias}^2 = \langle (\langle f_{\theta_0,\mathcal{D}}^*(\boldsymbol{x}) \rangle_{\theta_0,\mathcal{D}} - y)^2 \rangle_{\boldsymbol{x},y}, \tag{11}$$

$$V_\mathcal{D} = \mathbb{E}_{\boldsymbol{x}} \mathrm{Var}_\mathcal{D} \mathbb{E}_{\theta_0}[f_{\theta_0,\mathcal{D}}^*(\boldsymbol{x})|\mathcal{D}] = \mathbb{E}_{\boldsymbol{x}} \mathrm{Var}_\mathcal{D} \bar{f}_\mathcal{D}^*(\boldsymbol{x}), \tag{12}$$

$$V_{\theta_0} = \mathbb{E}_{\boldsymbol{x}} \mathrm{Var}_{\theta_0} \mathbb{E}_\mathcal{D}[f_{\theta_0,\mathcal{D}}^*(\boldsymbol{x})|\theta_0], \tag{13}$$

$$V_{\mathcal{D},\theta_0} = \mathbb{E}_{\boldsymbol{x}} \mathrm{Var}_{\theta_0,\mathcal{D}} f_{\theta_0,\mathcal{D}}^*(y) - V_{\theta_0} - V_\mathcal{D}. \tag{14}$$

$V_\mathcal{D}$ and $V_{\theta_0}$ give the components of the variance explained by variance in $\mathcal{D}, \theta_0$ respectively. $V_{\mathcal{D},\theta_0}$ is the remaining part of the variance not explained by either of these two sources. As in the main text, $\bar{f}_\mathcal{D}^*(\boldsymbol{x})$ is the ensemble average of the trained predictors over initializations. $\mathbb{E}_\mathcal{D}[f_{\theta_0,\mathcal{D}}^*(\boldsymbol{x})|\theta_0]$ is commonly referred to as the bagged predictor. In the next subsection we study these terms empirically.

## B.2 EMPIRICAL STUDY OF DATASET VARIANCE

Using the network simulations, one can show that the bagged predictor does not have substantially lower generalization error in the regimes that we are interested in. This implies that most of the variance driving higher generalization error is due to variance over initializations. In figure 7, we

make phase plots of the fraction of $E_g$ that arises from variance due to initialization, variance over datasets, and total variance for width 1000. This can be obtained by computing the ensembled predictor, the bagged predictor, and the ensembled-bagged predictor respectively.

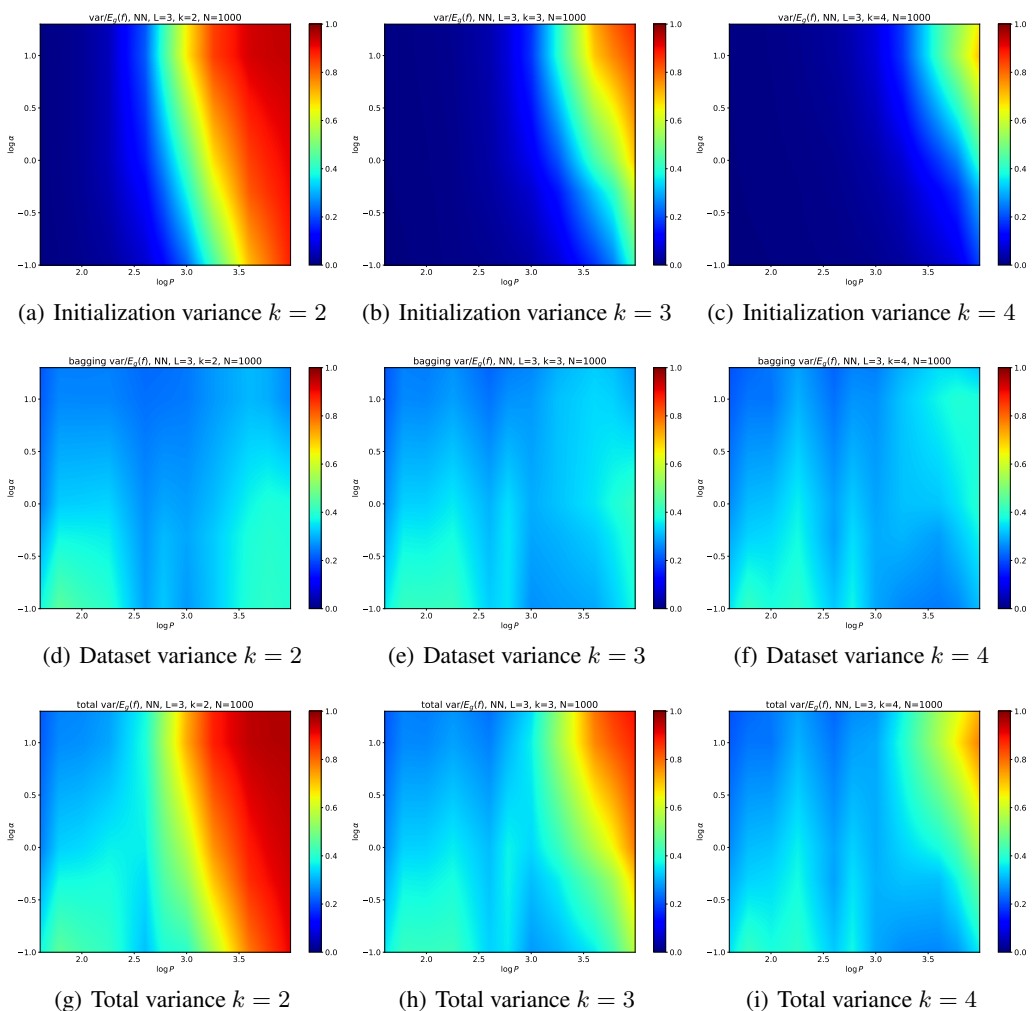

(a) Initialization variance $k = 2$    (b) Initialization variance $k = 3$    (c) Initialization variance $k = 4$

(d) Dataset variance $k = 2$    (e) Dataset variance $k = 3$    (f) Dataset variance $k = 4$

(g) Total variance $k = 2$    (h) Total variance $k = 3$    (i) Total variance $k = 4$

Figure 7: Phase plots of the fraction of the generalization error due to the initialization variance, the dataset variance, and their combined contribution. The columns correspond to the tasks of polynomial regression for degree $2, 3$ and $4$ polynomials. Neural network has width 1000 and depth 3. Notice that the initialization variance dominates in the large $P$ large $\alpha$ regime

### B.3 Relating Ensembled Network Generalization to Infinite Width Generalization

Making use of the fact that at leading order, the $\text{eNTK}_f$ (either in the rich or lazy regime) of a trained network has $\theta_0$-dependent fluctuations with variance $1/N$, one can write the kernel Gram matrices as

$$[\boldsymbol{K}_{\theta_0}]_{\mu\nu} = [\boldsymbol{K}_\infty]_{\mu\nu} + \frac{1}{\sqrt{N}}[\boldsymbol{\delta K}_{\theta_0}]_{\mu\nu} + O(1/N)$$

$$[\boldsymbol{k}_{\theta_0}(\boldsymbol{x})]_\mu = [\boldsymbol{k}_\infty(\boldsymbol{x})]_\mu + \frac{1}{\sqrt{N}}[\boldsymbol{\delta k}_{\theta_0}(\boldsymbol{x})]_\mu + O(1/N).$$

(15)

Here, $\boldsymbol{\delta K}_{\theta_0}, \boldsymbol{\delta k}_{\theta_0}$ are the leading order fluctuations around the infinite width network. Because of how we have written them, their variance is $O(1)$ with respect to $N$. Using perturbation theory

(Dyer & Gur-Ari, 2020), one can demonstrate that these leading order terms have mean zero around their infinite-width limit.

The predictor for the eNTK$_0$ (or for a sufficiently large $\alpha$ neural network) for a training set with target labels $\boldsymbol{y}$ is given by:

$$
\begin{aligned}
f^*(\boldsymbol{x})_{\theta_0} &= \boldsymbol{k}_{\theta_0}(\boldsymbol{x})_\mu^\top \boldsymbol{K}_{\theta_0}^{-1} \cdot \boldsymbol{y} \\
&= f^\infty(\boldsymbol{x}) + \frac{1}{\sqrt{N}} \boldsymbol{\delta k}_{\theta_0}(\boldsymbol{x})^\top \boldsymbol{K}_\infty^{-1} \cdot \boldsymbol{y} - \frac{1}{\sqrt{N}} \boldsymbol{k}_\infty(\boldsymbol{x})^\top \boldsymbol{K}_\infty^{-1} \boldsymbol{\delta K}_{\theta_0} \boldsymbol{K}_\infty^{-1} \cdot \boldsymbol{y} + O\left(N^{-1}\right).
\end{aligned}
$$
(16)

This implies (Geiger et al., 2020a):

$$
\langle (f_{\theta_0}^*(\boldsymbol{x}) - f^\infty(\boldsymbol{x}))^2 \rangle_{\boldsymbol{x}} = O\left(N^{-1}\right).
$$
(17)

Upon taking the ensemble, because of the mean zero property of the deviations, we get that

$$
\begin{aligned}
\langle f^*(\boldsymbol{x})_{\theta_0} \rangle_{\theta_0} &= f^\infty(\boldsymbol{x}) + O\left(N^{-1}\right) \\
&\Rightarrow \langle ((\langle f^*(\boldsymbol{x})_{\theta_0} \rangle_{\theta_0} - f^\infty(\boldsymbol{x}))^2 \rangle = O\left(N^{-2}\right).
\end{aligned}
$$
(18)

We can now bound the generalization error of the ensemble of networks in terms of the infinite-width generalization:

$$
\begin{aligned}
\langle ((\langle f_{\theta_0}^*(\boldsymbol{x}) \rangle_{\theta_0} - y)^2 \rangle_{\boldsymbol{x},y} &= \langle (f^\infty(\boldsymbol{x}) - y)^2 \rangle_{\boldsymbol{x},y} + \langle ((\langle f_{\theta_0}^*(\boldsymbol{x}) \rangle_{\theta_0} - f^\infty(\boldsymbol{x}))^2 \rangle_{\boldsymbol{x}} \\
&\quad - 2\langle (f^\infty(\boldsymbol{x}) - y)(f^\infty(\boldsymbol{x}) - \langle f_{\theta_0}^*(\boldsymbol{x}) \rangle_{\theta_0}) \rangle_{\boldsymbol{x},y}.
\end{aligned}
$$
(19)

By equation 18, the second term yields a positive contribution going as $O(N^{-2})$. The last term can be bounded by Cauchy-Schwarz:

$$
\begin{aligned}
|\langle (f^\infty(\boldsymbol{x}) - y)(f^\infty(\boldsymbol{x}) - \langle f_{\theta_0}^*(\boldsymbol{x}) \rangle_{\theta_0}) \rangle_{\boldsymbol{x},y}| &\leq \sqrt{\langle (f^\infty(\boldsymbol{x}) - y)^2 \rangle \langle (f^\infty(\boldsymbol{x}) - \langle f_{\theta_0}^*(\boldsymbol{x}) \rangle_{\theta_0})^2 \rangle_{\boldsymbol{x},y}} \\
&= \sqrt{E_g^\infty(P) c_1 / N}.
\end{aligned}
$$
(20)

After we enter the variance limited regime by taking $P > P_{1/2}$ we get $E_g^\infty \leq O(1/N)$ so this last term is bounded by $N^{-3/2}$. Consequently, the difference in generalization error between the infinite width NTK and an ensemble of lazy network or eNTK$_0$ predictors is subleading in $1/N$ compared to the generalization gap, which goes as $N^{-1}$.

The same argument can be extended to any predictor that differs from some infinite width limit. In particular Bordelon & Pehlevan (2022) show that the fluctuations of the eNTK$_f$ in any mean field network are asymptotically mean zero with variance $N^{-1}$. The above argument then applies to the predictor obtained by ensembling networks that have learned features. This implies that in the variance limited regime, ensemble averages of feature learning networks have the same generalization as the infinite-width mean field solutions up to a term that decays faster than $N^{-3/2}$.

## C  FEATURE LEARNING

### C.1  CONTROLLING FEATURE LEARNING THROUGH INITIALIZATION SCALE

Given the feed-forward network defined in equation 1, one can see that the components of the activations satisfy $h_i^{(\ell)} = O(\sigma h_i)^{(\ell-1)}$ and consequently that the output $h_1^{(L)} = O(\sigma^L)$. Because of the way the network is parameterized, the changes in the output $\frac{\partial f}{\partial \theta}$ also scale as $O(\sigma^L)$. This implies that the eNTK at any given time scales as

$$
K_\theta(\boldsymbol{x}, \boldsymbol{x}') = \sum_\theta \frac{\partial f(\boldsymbol{x})}{\partial \theta} \frac{\partial f(\boldsymbol{x}')}{\partial \theta} = O(\sigma^{2L}).
$$
(21)

After appropriately rescaling learning rate to $\eta = \sigma^{-2L}$ we get

$$
\frac{df(\boldsymbol{x})}{dt} = -\eta \sum_\mu K_\theta(\boldsymbol{x}, \boldsymbol{x}^\mu)(f(\boldsymbol{x}^\mu) - y^\mu).
$$
(22)

Under the assumption that $\sigma^L \ll 1$ and $y^\mu = O(1)$ so that the error term is $O(1)$ we get that the output changes in time as $df/dt = O(1)$.

On the other hand, using the chain rule one can show that the features change as a product of the gradient update and the features in the prior layer, yielding the scaling

$$\frac{dh^{(\ell)}}{dt} = \eta \frac{\sigma^L}{\sqrt{N}} = \frac{1}{\sigma^L \sqrt{N}} = (\alpha \sqrt{N})^{-1}. \tag{23}$$

This gives us that the change in the features scales as $(\alpha\sqrt{N})^{-1}$ while the change in the output scales as $O(1)$. Thus, for $\alpha\sqrt{N}$ sufficiently small, the features can move dramatically.

## C.2 OUTPUT RESCALING WITHOUT RESCALING WEIGHTS

In the main text, we use the scale $\sigma$ at every layer to change the scale of the output function. This relies on the homogeneity of the activation function so that $W^\ell \to \sigma W^\ell$ for all $\ell$ leads to a rescaling $f \to f\sigma^L$. This would not work for nonhomogenous activations like $\phi(h) = \tanh(h)$. However, following Chizat et al. (2019); Geiger et al. (2020a), we note that we can set all weights to be $O_\alpha(1)$ and introduce the $\alpha$ only in the definition of the neural network function

$$f = \frac{\alpha}{\sqrt{N}} \sum_{i=1}^N w_i^{L+1} \varphi(h_i^L) \,, \ h_i^\ell = \frac{1}{\sqrt{N}} \sum_{j=1}^N W_{ij}^\ell \varphi(h_j^{\ell-1}) \,, \ h_i^1 = \frac{1}{\sqrt{D}} W_{ij}^1 x_j. \tag{24}$$

We note that all preactivations $h^\ell$ have scale $O_\alpha(1)$ for any choice of nonlinearity, but that $f = \Theta_\alpha(\alpha)$. Several works have established that the $\alpha \sim \frac{1}{\sqrt{N}}$ allows feature learning even as the network approaches infinite width Mei et al. (2018); Yang & Hu (2020); Bordelon & Pehlevan (2022). This is known as the mean field or $\mu$-limit.

## C.3 KERNEL ALIGNMENT

In this section we comment on our choice of kernel alignment metric

$$A(\boldsymbol{K}) := \frac{\boldsymbol{y}^\top \boldsymbol{K} \boldsymbol{y}}{\mathrm{Tr}\, \boldsymbol{K} |\boldsymbol{y}|^2}. \tag{25}$$

For kernels that are diagonally dominant, such as those encountered in the experiments, this metric is related to another alignment metric

$$A_F(\boldsymbol{K}) := \frac{\boldsymbol{y}^\top \boldsymbol{K} \boldsymbol{y}}{|\boldsymbol{K}|_F |\boldsymbol{y}|^2}. \tag{26}$$

Here $|\boldsymbol{K}|_F$ is the Frobenius norm of the Gram matrix of the kernel. This metric was extensively used in Baratin et al. (2021). The advantage of the first metric over the second is that one can quickly estimate the denominator of $A(\boldsymbol{K})$ via Monte Carlo estimation of $\langle \boldsymbol{u}^\top \boldsymbol{K} \boldsymbol{u} \rangle_{\boldsymbol{u} \sim \mathcal{N}(0,\mathbf{1})}$.

We use $A(\boldsymbol{K}_f)$ as a measure of feature learning, as we have found that this more finely captures elements of feature learning than other related metrics. We list several metrics we tried that did not work.

One option for a representation-learning metric involves measuring the magnitude of the change between the initial and final kernels, $\boldsymbol{K}_i, \boldsymbol{K}_f$:

$$\Delta \boldsymbol{K} := |\boldsymbol{K}_f - \boldsymbol{K}_i|_F. \tag{27}$$

However, this is more sensitive to the raw parameter change than any task-relevant data. If one instead were to normalize the kernels to be unit norm at the beginning and the end, the modified metric

$$\Delta \boldsymbol{K} := \left| \frac{\boldsymbol{K}_f}{|\boldsymbol{K}_f|_F} - \frac{\boldsymbol{K}_i}{|\boldsymbol{K}_i|_F} \right|_F. \tag{28}$$

This metric however remains remarkably flat over the whole range of $\alpha, P$, as does the centered kernel alignment (CKA) of Cortes et al. (2012)

$$\mathrm{CKA}(\boldsymbol{K}_i, \boldsymbol{K}_f) = \frac{\mathrm{Tr}[\boldsymbol{K}_i^c \boldsymbol{K}_f^c]}{\sqrt{|\boldsymbol{K}_i^c|_F |\boldsymbol{K}_f^c|_F}}, \quad \boldsymbol{K}^c = \boldsymbol{C} \boldsymbol{K} \boldsymbol{C}, \quad \boldsymbol{C} = \mathbf{1} - \frac{1}{P} \vec{1} \vec{1}^{\,T}. \tag{29}$$

Here $C$ is the centering matrix that subtracts off the mean components of the kernel for a $P \times P$ kernel. This alignment metric has been shown to be useful in comparing neural representations (Kornblith et al., 2019). For our task, however, because the signal is low-dimensional, only a small set of eigenspaces of the kernel align to this task. As a result, the CKA, which counts all eigenspaces equally, appears to be too coarse to capture the low-dimensional feature learning that is happening.

On the other hand, we find that $A(\boldsymbol{K}_f)$ (with $\boldsymbol{K}_f$ given by the eNTK$_f$ evaluated on a test set) can very finely detect alignment along the task relevant directions. This produces a clear signal of feature learning at small $\alpha$ and large $P$ as shown in Figure 2c.

$A(\boldsymbol{K}_f)$ can be related to the centered kernel alignment between the eNTK$_f$ and the (mean zero) task kernel $\boldsymbol{y}\boldsymbol{y}^\top$, where $\boldsymbol{y}$ is a vector of draws from the population distribution $p(\boldsymbol{x}, y)$.

### C.4 RELATIONSHIP BETWEEN TRAINED NETWORK AND FINAL KERNEL

In general, the learned function contains contributions from the instantaneous NTKs at every point in the training. Concretely, following Atanasov et al. (2021) we have the following formula for the final network predictor $f(x)$

$$f(x) = \int_0^\infty dt\, \boldsymbol{k}(x, t) \cdot \exp\left(-\int_0^t ds\, \boldsymbol{K}(s)\right) \boldsymbol{y}, \tag{30}$$

where $[\boldsymbol{k}(x, t)]_\mu = K(x, x_\mu, t)$ and $[\boldsymbol{K}(s)]_{\mu\nu} = K(x_\mu, x_\nu, s)$ and $[\boldsymbol{y}]_\mu = y_\mu$. In general there are contributions from earlier kernels $\boldsymbol{k}(x, t)$ for $t < \infty$ and so the function $f$ cannot always be written as a linear combination of the final NTK $K_f$ on training data: $f = \sum_\mu \alpha_\mu K_f(x, x_\mu)$. However, as Vyas et al. (2022); Atanasov et al. (2021) have shown, the final predictions of the network are often well modeled by regression with the final NTK. We verify this for our task in section 3.2.

## D GENERIC RANDOM FEATURE MODEL

### D.1 SETTING UP THE PROBLEM: FEATURE DEFINITIONS

For a random kernel, $K(\boldsymbol{x}, \boldsymbol{x}'; \theta)$, we first compute its Mercer decomposition

$$\int d\boldsymbol{x}\, p(\boldsymbol{x}) K(\boldsymbol{x}, \boldsymbol{x}'; \theta) \phi_k(\boldsymbol{x}) = \lambda_k \phi_k(\boldsymbol{x}'). \tag{31}$$

From the eigenvalues $\lambda_k$ and eigenfunctions $\phi_k$, we can construct the square root

$$K^{1/2}(\boldsymbol{x}, \boldsymbol{x}'; \theta) = \sum_k \sqrt{\lambda_k} \phi_k(\boldsymbol{x}) \phi_k(\boldsymbol{x}'). \tag{32}$$

Lastly, using $K^{1/2}$, we can get a feature map by projecting against a static basis $\{b_k\}$ giving

$$\psi_k(\boldsymbol{x}) = \int d\boldsymbol{x}' p(\boldsymbol{x}') K^{1/2}(\boldsymbol{x}, \boldsymbol{x}'; \theta) b_k(\boldsymbol{x}'). \tag{33}$$

These features reproduce the kernel so that $K(\boldsymbol{x}, \boldsymbol{x}'; \theta) = \sum_k \psi_k(\boldsymbol{x}) \psi_k(\boldsymbol{x}')$. This can be observed from the following observation

$$\psi_k(\boldsymbol{x}) = \sum_\ell \sqrt{\lambda_\ell} \phi_\ell(\boldsymbol{x}) U_{\ell k}\,,\ U_{\ell k} = \langle \phi_\ell(\boldsymbol{x}) b_k(\boldsymbol{x})\rangle \tag{34}$$

$$\Rightarrow \sum_k \psi_k(\boldsymbol{x}) \psi_k(\boldsymbol{x}') = \sum_{\ell,m} \sqrt{\lambda_\ell \lambda_m} \phi_\ell(\boldsymbol{x}) \phi_m(\boldsymbol{x}') \sum_k U_{\ell k} U_{mk} = \sum_\ell \lambda_\ell \phi_\ell(\boldsymbol{x}), \phi_\ell(\boldsymbol{x}') \tag{35}$$

where the last line follows from the orthogonality of $U_{km}$ and recovers $K(\boldsymbol{x}, \boldsymbol{x}'; \theta)$.

### D.2 DECOMPOSITION OF FINITE WIDTH FEATURES

We now attempt to characterize the variance in the features over the sample distribution. We will first consider the case of a fixed realization of $\boldsymbol{\theta}_0$ before providing a typical case analysis over random $\boldsymbol{\theta}_0$. For a fixed initialization $\boldsymbol{\theta}_0$ we define the following covariance matrices

$$\boldsymbol{\Sigma}_M = \langle \boldsymbol{\psi}_M(\boldsymbol{x}) \boldsymbol{\psi}_M(\boldsymbol{x})^\top\rangle \in \mathbb{R}^{M \times M}. \tag{36}$$

where $\boldsymbol{\psi}_M$ are the truncated (but deterministic) features induced by the deterministic infinite width kernel. We will mainly be interested in the case where $M \to \infty$ and where the target function can be expressed as the linear combination $y(\boldsymbol{x}) = \boldsymbol{w}^* \cdot \boldsymbol{\psi}_M(\boldsymbol{x})$ of these features. For example, in the case of our experiments on the sphere, $\boldsymbol{\psi}_M$ could be the spherical harmonic functions. Further, in the $M \to \infty$ limit, we will be able to express the target features $\boldsymbol{\psi}$ as linear combinations of the features $\boldsymbol{\psi}_M$

$$\boldsymbol{\psi}(\boldsymbol{x}, \boldsymbol{\theta}_0) = \boldsymbol{A}(\boldsymbol{\theta}_0)\boldsymbol{\psi}_M(\boldsymbol{x}) \,, \;\; \boldsymbol{A}(\boldsymbol{\theta}) \in \mathbb{R}^{N_{\mathcal{H}} \times M}. \tag{37}$$

The matrix $\boldsymbol{A}(\boldsymbol{\theta}_0)$ are the coefficients of the decomposition which can vary over initializations. Crucially $\boldsymbol{A}(\boldsymbol{\theta}_0)$ projects to the subspace of dimension $N_{\mathcal{H}}$ where the finite width features have variance over $\boldsymbol{x}$. The population risk for this $\boldsymbol{\theta}_0$ has an irreducible component

$$E_g(\boldsymbol{\theta}_0) = \left\langle (\boldsymbol{w}^* \cdot \boldsymbol{\psi}_M - \boldsymbol{w} \cdot \boldsymbol{\psi})^2 \right\rangle$$
$$\geq \boldsymbol{w}^{*\top} \left[ \boldsymbol{\Sigma}_M - \boldsymbol{\Sigma}_M \boldsymbol{A}(\boldsymbol{\theta})^\top \left( \boldsymbol{A}(\boldsymbol{\theta}_0)\boldsymbol{\Sigma}_M \boldsymbol{A}(\boldsymbol{\theta}_0)^\top \right)^{-1} \boldsymbol{A}(\boldsymbol{\theta}_0)\boldsymbol{\Sigma}_M \right] \boldsymbol{w}^*. \tag{38}$$

where the bound is tight for the optimal weights $\boldsymbol{w} = \left( \boldsymbol{A}(\boldsymbol{\theta}_0)\boldsymbol{\Sigma}_M \boldsymbol{A}(\boldsymbol{\theta}_0)^\top \right)^{-1} \boldsymbol{A}(\boldsymbol{\theta}_0)\boldsymbol{\Sigma}_M \boldsymbol{w}^*$. The irreducible error is determined by a projection matrix which preserves the subspace where the features $\boldsymbol{\psi}(\boldsymbol{x}, \boldsymbol{\theta}_0)$ have variance: $\boldsymbol{I} - \boldsymbol{A}(\boldsymbol{\theta})^\top \left( \boldsymbol{A}(\boldsymbol{\theta}_0)\boldsymbol{\Sigma}_M \boldsymbol{A}(\boldsymbol{\theta}_0)^\top \right)^{-1} \boldsymbol{A}(\boldsymbol{\theta}_0)\boldsymbol{\Sigma}_M$. In general, this will preserve some fraction of the variance in the target function, but some variance in the target function will not be expressible by linear combinations of the features $\boldsymbol{\psi}(\boldsymbol{x}, \boldsymbol{\theta})$. We expect that random finite width $N$ neural networks will have unexplained variance in the target function on the order $\sim 1/N$.

### D.3 GAUSSIAN COVARIATE MODEL

Following prior works on learning curves for kernel regression (Bordelon et al., 2020; Canatar et al., 2021; Loureiro et al., 2021), we will approximate the learning problem with a Gaussian covariates model with matching second moments.

The features $\boldsymbol{\psi}_M(\boldsymbol{x})$ will be treated as Gaussian over random draws of datapoints. We will assume centered features. We decompose the features in the orthonormal basis $\boldsymbol{b}(\boldsymbol{x})$, which we approximate as a Gaussian vector $\boldsymbol{b} \sim \mathcal{N}(0, \boldsymbol{I})$.

$$f = \boldsymbol{\psi}(\boldsymbol{\theta}_0) \cdot \boldsymbol{w} \,, \;\; y = \bar{\boldsymbol{\psi}}_M \cdot \boldsymbol{w}^*$$
$$\boldsymbol{\psi}_M = \boldsymbol{\Sigma}_s^{1/2}\boldsymbol{b} \,, \;\; \boldsymbol{\psi}(\boldsymbol{\theta}_0) = \boldsymbol{A}(\boldsymbol{\theta}_0)^\top \boldsymbol{\psi}_M + \boldsymbol{\Sigma}_\epsilon^{1/2}\boldsymbol{\epsilon}$$
$$\boldsymbol{b} \sim \mathcal{N}(0, \boldsymbol{I}) \,, \;\; \boldsymbol{\epsilon} \sim \mathcal{N}(0, \boldsymbol{I}) \tag{39}$$

This is a special case of the Gaussian covariate model introduced by Loureiro et al. (2021) and subsumes the popular two-layer random feature models (Mei & Montanari, 2022; Adlam & Pennington, 2020b) as a special case. In a subsequent section, we go beyond Loureiro et al. (2021) by computing typical case learning curves over Gaussian $\boldsymbol{A}(\boldsymbol{\theta}_0)$ matrices. In particular, we have for the two layer random feature model in the proportional asymptotic limit $P, N, D \to \infty$ with $P/D = O(1)$ and $P/N = O(1)$ with $\boldsymbol{\psi}(\boldsymbol{x}) = \phi(\boldsymbol{F}\boldsymbol{x}_\mu)$ for fixed feature matrix $\boldsymbol{F} \in \mathbb{R}^{N \times D}$ nonlinearity $\phi$ and $\boldsymbol{x} = \boldsymbol{b} \sim \mathcal{N}(0, D^{-1}\boldsymbol{I})$

$$\boldsymbol{\Sigma}_M = \boldsymbol{I} \,, \;\; \boldsymbol{\Sigma}_\epsilon = c_*^2 \boldsymbol{I} \,, \;\; \boldsymbol{A}^\top = c_1 \boldsymbol{F}$$
$$c_1 = \langle z\phi(z) \rangle_{z \sim \mathcal{N}(0,1)} \,, \;\; c_*^2 = \langle \phi(z)^2 \rangle_{z \sim \mathcal{N}(0,1)} - c_1^2. \tag{40}$$

We refer readers to Hu & Lu (2020) for a discussion of this equivalence between random feature regression and this Gaussian covariate model.

### D.4 REPLICA CALCULATION OF THE LEARNING CURVE

To analyze the typical case performance of kernel regression, we define the following partition function:

$$Z[\mathcal{D}, \boldsymbol{\theta}_0] = \int d\boldsymbol{w} \exp \left( -\frac{\beta}{2\lambda} \sum_{\mu=1}^P [\boldsymbol{w} \cdot \boldsymbol{\psi}_\mu - \boldsymbol{w}^* \cdot \boldsymbol{\psi}_{M,\mu}]^2 - \frac{\beta}{2}|\boldsymbol{w}|^2 - \frac{J\beta M}{2} E_g(\boldsymbol{w}) \right)$$
$$E_g(\boldsymbol{w}) = \frac{1}{M}|\boldsymbol{\Sigma}_M^{1/2}\boldsymbol{w}^* - \boldsymbol{\Sigma}_M^{1/2}\boldsymbol{A}(\boldsymbol{\theta}_0)\boldsymbol{w}|^2 + \frac{1}{M}\boldsymbol{w}^\top \boldsymbol{\Sigma}_\epsilon \boldsymbol{w}. \tag{41}$$

For proper normalization, we assume that $\langle \boldsymbol{\psi}_M \boldsymbol{\psi}_M^\top \rangle = \frac{1}{M} \boldsymbol{\Sigma}_M$ and $\langle \boldsymbol{\epsilon} \boldsymbol{\epsilon}^\top \rangle = \frac{1}{M} \boldsymbol{\Sigma}_\epsilon$. We note that in the $\beta \to \infty$ limit, the partition function is dominated by the unique minimizer of the regularized least squares objective (Canatar et al., 2021; Loureiro et al., 2021). Further, for a fixed realization of $\boldsymbol{\theta}_0$ the average generalization error over datasets $\mathcal{D}$ can be computed by differentiation of the source term $J$

$$
\frac{2}{\beta M} \frac{\partial}{\partial J} |_{J=0} \langle \ln Z[\mathcal{D}, \boldsymbol{\theta}_0] \rangle_{\mathcal{D}}
$$
$$
= \left\langle \frac{1}{Z} \int d\boldsymbol{w} \exp \left( -\frac{\beta}{2\lambda} \sum_{\mu=1}^{P} [\boldsymbol{w} \cdot \boldsymbol{\psi}_\mu - \boldsymbol{w}^* \cdot \boldsymbol{\psi}_{M,\mu}]^2 - \frac{\beta}{2} |\boldsymbol{w}|^2 \right) E_g(\boldsymbol{w}) \right\rangle_{\mathcal{D}} . \tag{42}
$$

Thus the $\beta \to \infty$ limit of the above quantity will give the expected generalization error of the risk minimizer. We see the need to average the quantity $\ln Z$ over realizations of datasets $\mathcal{D}$. For this, we resort to the replica trick $\langle \ln Z \rangle = \lim_{n \to 0} \frac{1}{n} \ln \langle Z^n \rangle$. We will compute the integer moments $\langle Z^n \rangle$ for integer $n$ and then analytically continue the resulting expressions to $n \to 0$ under a symmetry ansatz. The replicated partition function thus has the form

$$
\langle Z^n \rangle = \int \prod_{a=1}^{n} d\boldsymbol{w}^a \mathbb{E}_{\{\boldsymbol{b}_\mu, \boldsymbol{\epsilon}_\mu\}} \exp \left( -\frac{\beta}{2\lambda} \sum_{\mu=1}^{P} \sum_{a=1}^{n} [\boldsymbol{w}^a \cdot \boldsymbol{\psi}_\mu - \boldsymbol{w}^* \cdot \boldsymbol{\psi}_{M,\mu}]^2 - \frac{\beta}{2} \sum_{a=1}^{n} |\boldsymbol{w}^a|^2 \right)
$$
$$
\times \exp \left( -\frac{J\beta M}{2} \sum_{a=1}^{n} E_g(\boldsymbol{w}^a) \right) . \tag{43}
$$

We now need to perform the necessary average over the random realizations of data points $\mathcal{D} = \{\boldsymbol{b}_\mu, \boldsymbol{\epsilon}_\mu\}$. We note that the scalar quantities $h_\mu^a = \boldsymbol{w}^a \cdot \boldsymbol{\psi}_\mu - \boldsymbol{w}^* \cdot \boldsymbol{\psi}_{M,\mu}$ are Gaussian with mean zero and covariance

$$
\langle h_\mu^a h_\nu^b \rangle = \delta_{\mu\nu} Q_{ab}
$$
$$
Q_{ab} = \frac{1}{M} \left( \boldsymbol{A}(\boldsymbol{\theta}_0) \boldsymbol{w}^a - \boldsymbol{w}^* \right) \boldsymbol{\Sigma}_M \left( \boldsymbol{A}(\boldsymbol{\theta}_0) \boldsymbol{w}^a - \boldsymbol{w}^* \right) + \frac{1}{M} \boldsymbol{w}^a \boldsymbol{\Sigma}_\epsilon \boldsymbol{w}^b . \tag{44}
$$

We further see that the generalization error in replica $a$ is $E_g(\boldsymbol{w}^a) = Q_{aa}$. Performing the Gaussian integral over $\{h_\mu^a\}$ gives

$$
\langle Z^n \rangle \propto \int \prod_a d\boldsymbol{w}^a \prod_{ab} dQ_{ab} d\hat{Q}_{ab} \exp \left( -\frac{P}{2} \ln \det (\lambda \boldsymbol{I} + \beta \boldsymbol{Q}) - \frac{J\beta M}{2} \text{Tr} \boldsymbol{Q} - \frac{\beta}{2} \sum_a |\boldsymbol{w}^a|^2 \right)
$$
$$
\exp \left( \frac{1}{2} \sum_{ab} \hat{Q}_{ab} \left( M Q_{ab} - [\boldsymbol{A}(\boldsymbol{\theta}_0) \boldsymbol{w}^a - \boldsymbol{w}^*]^\top \boldsymbol{\Sigma}_M [\boldsymbol{A}(\boldsymbol{\theta}_0) \boldsymbol{w}^a - \boldsymbol{w}^*] + \boldsymbol{w}^a \boldsymbol{\Sigma}_\epsilon \boldsymbol{w}^b \right) \right) .
$$

We introduced the Lagrange multipliers $\hat{\boldsymbol{Q}}$ which enforce the definition of order parameters $\boldsymbol{Q}$. We now integrate over $\boldsymbol{W} = \text{Vec}\{\boldsymbol{w}^a\}_{a=1}^n$. We let $\tilde{\boldsymbol{\Sigma}}_s = \boldsymbol{A}^\top \boldsymbol{\Sigma}_M \boldsymbol{A}$

$$
\int d\boldsymbol{W} \exp \left( -\frac{1}{2} \boldsymbol{W} \left[ \beta \boldsymbol{I} + \hat{\boldsymbol{Q}} \otimes [\tilde{\boldsymbol{\Sigma}}_s + \boldsymbol{\Sigma}_\epsilon] \right] \boldsymbol{W} \right)
$$
$$
\exp \left( \boldsymbol{W}^\top \left[ \hat{\boldsymbol{Q}} \otimes \boldsymbol{I} \right] \left( \boldsymbol{1} \otimes \boldsymbol{A}^\top \boldsymbol{\Sigma}_s \boldsymbol{w}^* \right) \right)
$$
$$
= \exp \left( \frac{1}{2} \left( \boldsymbol{1} \otimes \boldsymbol{A}^\top \boldsymbol{\Sigma}_s \boldsymbol{w}^* \right)^\top \left[ \hat{\boldsymbol{Q}} \otimes \boldsymbol{I} \right] \left[ \beta \boldsymbol{I} + \hat{\boldsymbol{Q}} \otimes [\tilde{\boldsymbol{\Sigma}}_s + \boldsymbol{\Sigma}_\epsilon] \right]^{-1} \left[ \hat{\boldsymbol{Q}} \otimes \boldsymbol{I} \right] \left( \boldsymbol{1} \otimes \boldsymbol{A}^\top \boldsymbol{\Sigma}_s \boldsymbol{w}^* \right) \right)
$$
$$
\exp \left( -\frac{1}{2} \ln \det \left[ \beta \boldsymbol{I} + \hat{\boldsymbol{Q}} \otimes [\tilde{\boldsymbol{\Sigma}}_s + \boldsymbol{\Sigma}_\epsilon] \right] \right) . \tag{45}
$$

To take the $n \to 0$ limit, we make the replica symmetry ansatz

$$
\beta \boldsymbol{Q} = q \boldsymbol{I} + q_0 \boldsymbol{1} \boldsymbol{1}^\top , \ \beta^{-1} \hat{\boldsymbol{Q}} = \hat{q} \boldsymbol{I} + \hat{q}_0 \boldsymbol{1} \boldsymbol{1}^\top , \tag{46}
$$

which is well motivated since this is a convex optimization problem. Letting $\alpha = P/N$, we find that under the RS ansatz the replicated partition function has the form

$$\langle Z^n \rangle = \int dq \, dq_0 \, d\hat{q} \, d\hat{q}_0 \exp\left(\frac{nM}{2} S[q, q_0, \hat{q}, \hat{q}_0]\right) \tag{47}$$

$$S = q\hat{q} + q_0\hat{q} + q\hat{q}_0 - \alpha\left[\ln(\lambda + q) + \frac{q_0}{\lambda + q}\right]$$

$$- \frac{\beta}{M} \boldsymbol{w}^*[\hat{q}\boldsymbol{\Sigma}_M]\boldsymbol{w}^* + \frac{\beta}{M}\boldsymbol{w}^*[\hat{q}\boldsymbol{\Sigma}_s]\boldsymbol{A}^\top \boldsymbol{G}\boldsymbol{A}[\hat{q}\boldsymbol{\Sigma}_s]\boldsymbol{w}^*$$

$$+ \frac{1}{M}\ln\det\boldsymbol{G} - \frac{1}{M}\hat{q}_0 \text{Tr}\boldsymbol{G}[\tilde{\boldsymbol{\Sigma}}_s + \boldsymbol{\Sigma}_\epsilon] - J(q + q_0)$$

$$\boldsymbol{G} = \left(\boldsymbol{I} + \hat{q}[\tilde{\boldsymbol{\Sigma}}_s + \boldsymbol{\Sigma}_\epsilon]\right)^{-1}. \tag{48}$$

In a limit where $\alpha = P/M$ is $O(1)$, then this $S$ is intensive $S = O_M(1)$. We can thus appeal to saddle point integration (method of steepest descent) to compute the set of order parameters which have dominant contribution to the free energy.

$$\langle Z^n \rangle = \int dq \, dq_0 \, d\hat{q} \, d\hat{q}_0 \exp\left(\frac{nM}{2} S[q, q_0, \hat{q}, \hat{q}_0]\right) \sim \exp\left(\frac{nM}{2} S[q^*, q_0^*, \hat{q}^*, \hat{q}_0^*]\right)$$

$$\implies \langle \ln Z \rangle = \frac{M}{2} S[q^*, q_0^*, \hat{q}^*, \hat{q}_0^*]. \tag{49}$$

The order parameters $q^*, q_0^*, \hat{q}^*, \hat{q}_0^*$ are defined via the saddle point equations $\frac{\partial S}{\partial q} = \frac{\partial S}{\partial q_0} = \frac{\partial S}{\partial \hat{q}} = \frac{\partial S}{\partial \hat{q}_0} = 0$. For our purposes, it suffices to analyze two of these equations

$$\frac{\partial S}{\partial q_0} = \hat{q} - \frac{\alpha}{\lambda + q} - J = 0,$$

$$\frac{\partial S}{\partial \hat{q}_0} = q - \frac{1}{M}\text{Tr}\boldsymbol{G}\left[\tilde{\boldsymbol{\Sigma}}_s + \boldsymbol{\Sigma}_\epsilon\right] = 0. \tag{50}$$

We can now take the zero temperature ($\beta \to \infty$) limit to solve for the generalization error

$$E_g = -\frac{\partial}{\partial J}|_{J=0} \lim_{\beta \to \infty} \frac{1}{\beta} F$$

$$= \frac{1}{M}\partial_J \boldsymbol{w}^* \left[\hat{q}\boldsymbol{\Sigma}_M - \hat{q}^2 \boldsymbol{\Sigma}_M \boldsymbol{A}^\top \boldsymbol{G}\boldsymbol{A}\boldsymbol{\Sigma}_M\right]\boldsymbol{w}^*. \tag{51}$$

We see that we need to compute the $J$ derivatives on $\hat{q}$. We let $\kappa = \lambda + q$ and note

$$\partial_J \hat{q} = -\alpha\kappa^{-2}\partial_J \kappa + 1$$

$$\partial_J \kappa = -\partial_J \hat{q} \frac{1}{M}\text{Tr}\boldsymbol{G}^2\left[\tilde{\boldsymbol{\Sigma}}_s + \boldsymbol{\Sigma}_\epsilon\right]^2 = -\left(-\alpha\kappa^{-2}\partial_J\kappa + 1\right)\frac{1}{M}\text{Tr}\boldsymbol{G}^2\left[\tilde{\boldsymbol{\Sigma}}_s + \boldsymbol{\Sigma}_\epsilon\right]^2. \tag{52}$$

We solve the equation for $\partial_J \kappa$ which gives $\partial_J \kappa = -\frac{\kappa^2}{\alpha}\frac{\gamma}{1-\gamma}$ where $\gamma = \frac{\alpha}{\kappa^2}\frac{1}{M}\text{Tr}\boldsymbol{G}^2\left[\tilde{\boldsymbol{\Sigma}}_s + \boldsymbol{\Sigma}_\epsilon\right]^2$. With this definition we have $\partial_J \hat{q} = 1 + \frac{\gamma}{1-\gamma} = \frac{1}{1-\gamma}$.

$$E_g = \frac{1}{1-\gamma}\frac{1}{M}\boldsymbol{w}^*\boldsymbol{\Sigma}_M^{1/2}\left[\boldsymbol{I} - 2\hat{q}\boldsymbol{\Sigma}_s^{1/2}\boldsymbol{A}^\top\boldsymbol{G}\boldsymbol{A}\boldsymbol{\Sigma}_s^{1/2} + \hat{q}^2\boldsymbol{\Sigma}_s^{1/2}\boldsymbol{A}^\top\boldsymbol{G}\left[\tilde{\boldsymbol{\Sigma}}_s + \boldsymbol{\Sigma}_\epsilon\right]\boldsymbol{G}\boldsymbol{A}\boldsymbol{\Sigma}_s^{1/2}\right]\boldsymbol{\Sigma}_M^{1/2}\boldsymbol{w}^*$$

$$= \frac{1}{1-\gamma}\frac{1}{M}\boldsymbol{w}^*\boldsymbol{\Sigma}_M^{1/2}\left[\boldsymbol{I} - \hat{q}\boldsymbol{\Sigma}_s^{1/2}\boldsymbol{A}^\top\boldsymbol{G}\boldsymbol{A}\boldsymbol{\Sigma}_s^{1/2} - \hat{q}\boldsymbol{\Sigma}_s^{1/2}\boldsymbol{A}^\top\boldsymbol{G}^2\boldsymbol{A}\boldsymbol{\Sigma}_s^{1/2}\right]\boldsymbol{\Sigma}_M^{1/2}\boldsymbol{w}^*. \tag{53}$$

This reproduces the derived expression from Loureiro et al. (2021). The matching covariance $\tilde{\boldsymbol{\Sigma}}_s = \boldsymbol{\Sigma}_M$ and zero feature-noise limit $\boldsymbol{\Sigma}_\epsilon = 0$ recovers the prior results of Bordelon et al. (2020); Canatar et al. (2021); Simon et al. (2021). In general, this error will asymptote to the the irreducible error

$$\lim_{P \to \infty} E_g = \frac{1}{M}\boldsymbol{w}^*\left[\boldsymbol{\Sigma}_M - \boldsymbol{\Sigma}_s\boldsymbol{A}^\top\left(\tilde{\boldsymbol{\Sigma}}_s + \boldsymbol{\Sigma}_\epsilon\right)^{-1}\boldsymbol{A}\boldsymbol{\Sigma}_s\right]\boldsymbol{w}^*. \tag{54}$$

We see that this recovers the minimal possible error in the $P \to \infty$ limit. The derived learning curves depend on the instance of random initial condition $\boldsymbol{\theta}_0$. To get the average case performance, we take an additional average of this expression over $\boldsymbol{\theta}_0$

$$\mathbb{E}_{\boldsymbol{\theta}_0} E_g(\boldsymbol{\theta}_0) = \mathbb{E}_{\theta_0} \frac{1}{1-\gamma} \frac{1}{M} \boldsymbol{w}^* \boldsymbol{\Sigma}_M^{1/2} \left[ \boldsymbol{I} - \hat{q} \boldsymbol{\Sigma}_s^{1/2} \boldsymbol{A}^\top \boldsymbol{G} \boldsymbol{A} \boldsymbol{\Sigma}_s^{1/2} - \hat{q} \boldsymbol{\Sigma}_s^{1/2} \boldsymbol{A}^\top \boldsymbol{G}^2 \boldsymbol{A} \boldsymbol{\Sigma}_s^{1/2} \right] \boldsymbol{\Sigma}_M^{1/2} \boldsymbol{w}^*. \tag{55}$$

This average is complicated since $\gamma, \hat{q}, \boldsymbol{G}$ all depend on $\boldsymbol{\theta}_0$. In the next section we go beyond this analysis to try average case analysis for random Gaussian $\boldsymbol{A}$.

### D.5 QUENCHED AVERAGE OVER GAUSSIAN A

In this section we will define a distribution of features which allows an exact asymptotic prediction over random realizations of disorder $\boldsymbol{\theta}_0$ and datasets $\mathcal{D}$. This is a nontrivial extension of the result of Loureiro et al. (2021) since the number of necessary saddle point equations to be solved doubles from two to four. However, this more complicated theory allows us to exactly compute the expectation in equation 55 under an ansatz for the random matrix $\boldsymbol{A}$. We construct our features with

$$\boldsymbol{\psi} | \boldsymbol{A} = \frac{1}{\sqrt{N}} \boldsymbol{A}^\top \boldsymbol{\psi}_M + \boldsymbol{\Sigma}_\epsilon^{1/2} \boldsymbol{\epsilon} \, , \ A_{ij} \sim \mathcal{N}(0, \sigma^2).$$

We will now perform an approximate average over both datasets $\mathcal{D}$ and realizations of $\boldsymbol{A}$

$$\langle Z^n \rangle = \int \prod_{a=1}^n d\boldsymbol{w}^a \mathbb{E}_{\{\boldsymbol{b}_\mu, \boldsymbol{\epsilon}_\mu, \boldsymbol{A}\}} \exp \left( -\frac{\beta}{2\lambda} \sum_{\mu=1}^P \sum_{a=1}^n [\boldsymbol{w}^a \cdot \boldsymbol{\psi}_\mu - \boldsymbol{w}^* \cdot \boldsymbol{\psi}_{M,\mu}]^2 - \frac{\beta}{2} \sum_{a=1}^n |\boldsymbol{w}^a|^2 \right)$$

$$\times \exp \left( -\frac{JM\beta}{2} \sum_{a=1}^n E_g(\boldsymbol{w}^a) \right). \tag{56}$$

As before, we first average over $\boldsymbol{b}_\mu, \boldsymbol{\epsilon}_\mu | A$ and define order parameters $Q_{ab}$ as before.

$$\langle Z^n \rangle = \int \prod_a d\boldsymbol{w}^a \prod_{ab} dQ_{ab} d\hat{Q}_{ab} \exp \left( -\frac{P}{2} \ln \det (\lambda \boldsymbol{I} + \beta \boldsymbol{Q}) - \frac{J\beta M}{2} \text{Tr} \boldsymbol{Q} - \frac{\beta}{2} \sum_a |\boldsymbol{w}^a|^2 \right)$$

$$\mathbb{E}_{\{\boldsymbol{g}^a\}} \exp \left( \frac{1}{2} \sum_{ab} \hat{Q}_{ab} \left( M Q_{ab} - [\boldsymbol{g}^a - \boldsymbol{w}^*]^\top \boldsymbol{\Sigma}_M [\boldsymbol{g}^b - \boldsymbol{w}^*] + \boldsymbol{w}^a \boldsymbol{\Sigma}_\epsilon \boldsymbol{w}^b \right) \right).$$

where we defined the fields $\boldsymbol{g}^a = \frac{1}{\sqrt{N}} \boldsymbol{A} \boldsymbol{w}^a$ which are mean zero Gaussian with covariance $\langle \boldsymbol{g}^a \boldsymbol{g}^{b\top} \rangle = V_{ab} \boldsymbol{I}$ where $V_{ab} = \frac{\sigma^2}{N} \boldsymbol{w}^a \cdot \boldsymbol{w}^b$. Performing the Gaussian integral over $\boldsymbol{G} = \text{Vec}\{\boldsymbol{g}^a\}$, we find

$$\int \prod_a d\boldsymbol{g}^a \exp \left( -\frac{1}{2} \boldsymbol{G} \left[ \boldsymbol{I} \otimes \boldsymbol{V}^{-1} + \boldsymbol{\Sigma}_M \otimes \hat{\boldsymbol{Q}} \right] \boldsymbol{G} + (\boldsymbol{\Sigma}_M \boldsymbol{w}^* \otimes \hat{\boldsymbol{Q}} \mathbf{1}) \boldsymbol{G} - \frac{1}{2} \ln \det (\boldsymbol{I} \otimes \boldsymbol{V}) \right)$$

$$= \exp \left( \frac{1}{2} (\boldsymbol{\Sigma}_M \boldsymbol{w}^* \otimes \hat{\boldsymbol{Q}} \mathbf{1}) \left[ \boldsymbol{I} \otimes \boldsymbol{V}^{-1} + \boldsymbol{\Sigma}_M \otimes \hat{\boldsymbol{Q}} \right]^{-1} (\boldsymbol{\Sigma}_M \boldsymbol{w}^* \otimes \hat{\boldsymbol{Q}} \mathbf{1}) - \frac{1}{2} \ln \det \left( \boldsymbol{I} + \boldsymbol{\Sigma}_M \otimes \hat{\boldsymbol{Q}} \boldsymbol{V} \right) \right). \tag{57}$$

Next, we need to integrate over $\boldsymbol{W} = \text{Vec}\{\boldsymbol{w}^a\}$ which gives

$$\int d\boldsymbol{W} \exp \left( -\frac{1}{2} \boldsymbol{W} \left[ \beta \boldsymbol{I} + \sigma^2 \boldsymbol{I} \otimes \hat{\boldsymbol{V}} + \boldsymbol{\Sigma}_\epsilon \otimes \hat{\boldsymbol{Q}} \right] \boldsymbol{W} \right) = \exp \left( -\frac{1}{2} \ln \det \left[ \beta \boldsymbol{I} + \sigma^2 \boldsymbol{I} \otimes \hat{\boldsymbol{V}} + \boldsymbol{\Sigma}_\epsilon \otimes \hat{\boldsymbol{Q}} \right] \right). \tag{58}$$

Now the replicated partition function has the form

$$
\langle Z^n \rangle = \int d\boldsymbol{Q} d\hat{\boldsymbol{Q}} d\boldsymbol{V} d\hat{\boldsymbol{V}} \exp\left( \frac{M}{2} \mathrm{Tr}[\boldsymbol{Q}\hat{\boldsymbol{Q}} + \eta \boldsymbol{V}\hat{\boldsymbol{V}}] - \frac{J\beta M}{2} \mathrm{Tr}\boldsymbol{Q} - \frac{P}{2} \ln \det\left[ \lambda \boldsymbol{I} + \beta \boldsymbol{Q} \right] \right)
$$
$$
\times \exp\left( -\frac{1}{2}(\boldsymbol{w}^* \otimes \mathbf{1})^\top [\boldsymbol{\Sigma}_M \otimes \hat{\boldsymbol{Q}}](\boldsymbol{w}^* \otimes \mathbf{1}) \right)
$$
$$
\times \exp\left( \frac{1}{2}(\boldsymbol{\Sigma}_M \boldsymbol{w}^* \otimes \hat{\boldsymbol{Q}}\mathbf{1})^\top \left[ \boldsymbol{I} \otimes \boldsymbol{V}^{-1} + \boldsymbol{\Sigma}_M \otimes \hat{\boldsymbol{Q}} \right]^{-1} (\boldsymbol{\Sigma}_M \boldsymbol{w}^* \otimes \hat{\boldsymbol{Q}}\mathbf{1}) \right) \tag{59}
$$
$$
\times \exp\left( -\frac{1}{2} \ln \det\left[ \boldsymbol{I} + \boldsymbol{\Sigma}_M \otimes \hat{\boldsymbol{Q}}\boldsymbol{V} \right] - \frac{1}{2} \ln \det\left[ \beta \boldsymbol{I} + \sigma^2 \boldsymbol{I} \otimes \hat{\boldsymbol{V}} + \boldsymbol{\Sigma}_\epsilon \otimes \hat{\boldsymbol{Q}} \right] \right).
$$

Now we make a replica symmetry ansatz on the order parameters $\boldsymbol{Q}, \hat{\boldsymbol{Q}}, \boldsymbol{V}, \hat{\boldsymbol{V}}$

$$
\beta \boldsymbol{Q} = q\boldsymbol{I} + q_0 \mathbf{1}\mathbf{1}^\top \,,\ \beta \boldsymbol{V} = v\boldsymbol{I} + v_0 \mathbf{1}\mathbf{1}^\top
$$
$$
\beta^{-1}\hat{\boldsymbol{Q}} = \hat{q}\boldsymbol{I} + \hat{q}_0 \mathbf{1}\mathbf{1}^\top \,,\ \beta^{-1}\hat{\boldsymbol{V}} = \hat{v}\boldsymbol{I} + \hat{v}_0 \mathbf{1}\mathbf{1}^\top. \tag{60}
$$

We introduce the shorthand for normalized trace of a matrix $\boldsymbol{G}$ as $\mathrm{tr}\,\boldsymbol{G} = \frac{1}{M}\mathrm{Tr}\boldsymbol{G}$. Under the replica symmetry ansatz, we find the following free energy

$$
\frac{2}{M} \langle \ln Z \rangle = q\hat{q} + q_0\hat{q} + q\hat{q}_0 + \eta(v\hat{v} + v_0\hat{v} + v\hat{v}_0) - J(q + q_0) - \alpha \left[ \ln(\lambda + q) + \frac{q_0}{\lambda + q} \right]
$$
$$
- \frac{\beta}{M} \boldsymbol{w}^* [\hat{q}\boldsymbol{\Sigma}_M] \boldsymbol{w}^* + \frac{\beta}{M} \boldsymbol{w}^* [\hat{q}\boldsymbol{\Sigma}_M][v^{-1}\boldsymbol{I} + \hat{q}\boldsymbol{\Sigma}_M]^{-1}[\hat{q}\boldsymbol{\Sigma}_M]
$$
$$
- \mathrm{tr}\log\left[ \boldsymbol{I} + \hat{q}v\boldsymbol{\Sigma}_M \right] - (\hat{q}_0 v + \hat{q}v_0)\,\mathrm{tr}[\boldsymbol{I} + \hat{q}v\boldsymbol{\Sigma}_M]^{-1}\boldsymbol{\Sigma}_M
$$
$$
- \mathrm{tr}\log\left[ \boldsymbol{I} + \sigma^2\hat{v}\boldsymbol{I} + \boldsymbol{\Sigma}_\epsilon\hat{q} \right] - \mathrm{tr}\left[ \boldsymbol{I} + \sigma^2\hat{v}\boldsymbol{I} + \boldsymbol{\Sigma}_\epsilon\hat{q} \right]^{-1} \left[ \hat{v}_0\sigma^2\boldsymbol{I} + \hat{q}_0\boldsymbol{\Sigma}_\epsilon \right]. \tag{61}
$$

Letting $F = 2M^{-1}\langle \ln Z \rangle$, the saddle point equations read

$$
\frac{\partial F}{\partial q_0} = \hat{q} - \frac{\alpha}{\lambda + q} - J = 0,
$$
$$
\frac{\partial F}{\partial \hat{q}_0} = q - v\,\mathrm{tr}[\boldsymbol{I} + \hat{q}v\boldsymbol{\Sigma}_M]^{-1}\boldsymbol{\Sigma}_M - \mathrm{tr}[\boldsymbol{I} + \sigma^2\hat{v}\boldsymbol{I} + \boldsymbol{\Sigma}_\epsilon\hat{q}]^{-1}\boldsymbol{\Sigma}_\epsilon = 0,
$$
$$
\frac{\partial F}{\partial v_0} = \eta\hat{v} - \hat{q}\,\mathrm{tr}[\boldsymbol{I} + \hat{q}v\boldsymbol{\Sigma}_M]^{-1}\boldsymbol{\Sigma}_M = 0,
$$
$$
\frac{\partial F}{\partial \hat{v}_0} = \eta v - \sigma^2\,\mathrm{tr}[\boldsymbol{I} + \sigma^2\hat{v}\boldsymbol{I} + \boldsymbol{\Sigma}_\epsilon\hat{q}]^{-1} = 0. \tag{62}
$$

Now the generalization error can be determined from

$$
E_g = -\frac{\partial}{\partial J}\lim_{\beta\to\infty}\frac{2}{\beta M}\langle \ln Z \rangle = \partial_J \frac{1}{M}\boldsymbol{w}^*\left[ \hat{q}\boldsymbol{\Sigma}_M - (\hat{q}\boldsymbol{\Sigma}_M)[v^{-1}\boldsymbol{I} + \hat{q}\boldsymbol{\Sigma}_M]^{-1}(\hat{q}\boldsymbol{\Sigma}_M) \right]\boldsymbol{w}^*. \tag{63}
$$

We see that it is necessary to compute $\partial_J \hat{q}$ and $\partial_J v$ in order to obtain the final result. For simplicity, we set $\sigma^2 = 1$. The equations for the source derivatives are

$$
\partial_J \hat{q} = -\frac{\alpha}{(\lambda + q)^2}\partial_J q + 1,
$$
$$
\partial_J q = -\mathrm{tr}[\boldsymbol{I} + v\hat{q}\boldsymbol{\Sigma}_M]^{-2}[-\partial_J v\boldsymbol{I} + v^2\partial_J\hat{q}\boldsymbol{\Sigma}_M]\boldsymbol{\Sigma}_M - \mathrm{tr}[\boldsymbol{I} + \hat{v}\boldsymbol{I} + \hat{q}\boldsymbol{\Sigma}_\epsilon]^{-2}\boldsymbol{\Sigma}_\epsilon[\partial_J\hat{v}\boldsymbol{I} + \partial_J\hat{q}\boldsymbol{\Sigma}_\epsilon],
$$
$$
\eta\partial_J\hat{v} = -\mathrm{tr}[\boldsymbol{I} + v\hat{q}\boldsymbol{\Sigma}_M]^{-2}\boldsymbol{\Sigma}_M[-\partial_J\hat{q}\boldsymbol{I} + \hat{q}^2\partial_J v\boldsymbol{\Sigma}_M],
$$
$$
\eta\partial_J v = -\mathrm{tr}[\boldsymbol{I} + \hat{v}\boldsymbol{I} + \boldsymbol{\Sigma}_\epsilon\hat{q}]^{-2}[\partial_J\hat{v}\boldsymbol{I} + \partial_J\hat{q}\boldsymbol{\Sigma}_\epsilon]. \tag{64}
$$

Once the value of the order parameters $(q, \hat{q}, v, \hat{v})$ have been determined, these source derivatives can be obtained by solving a $4 \times 4$ linear system. Examples of these solutions are provided in Figure 16.

### D.5.1 Asymptotics in Underparameterized Regime

We can compute the asymptotic ($\alpha \to \infty$) generalization error due to the random projection $\boldsymbol{A}$ in the limit of $\boldsymbol{\Sigma}_\epsilon = 0$. First, note that if $\hat{v} \to O_\alpha(1)$, then the asymptotic error would be zero. Therefore, we will assume that $\hat{v} \sim a\alpha^c$ for some $a, c > 0$. The saddle point equations give the following asymptotic conditions

$$\hat{q} \sim \frac{\alpha}{\lambda} \ , \ \eta \sim \hat{q} \operatorname{tr}[\hat{v}\boldsymbol{I} + \hat{q}\boldsymbol{\Sigma}_M]^{-1}\boldsymbol{\Sigma}_M$$
$$\implies \eta = \operatorname{tr}[\lambda a\alpha^{c-1}\boldsymbol{I} + \boldsymbol{\Sigma}_M]^{-1}\boldsymbol{\Sigma}_M. \tag{65}$$

For $0 < \eta < 1$, this equation can only be satisfied as $\alpha \to \infty$ if $c = 1$ so that $\hat{v}$ has the same scaling with $\alpha$ as $\hat{q}$. If $c < 1$ then we could get the equation $\eta = 1$. If $c > 1$, then the equation would give $\eta = 0$. The constant $a$ solves the equation

$$\eta = \operatorname{tr}[\lambda a\boldsymbol{I} + \boldsymbol{\Sigma}_M]^{-1}\boldsymbol{\Sigma}_M. \tag{66}$$

Using this fact, our order parameters satisfy the following large $\alpha$ scalings

$$\hat{q} \sim \frac{\alpha}{\lambda} \ , \ q \sim 0 \ , \ \hat{v} \sim a\alpha \ , \ v \sim 0. \tag{67}$$

The source derivative equations simplify to $\partial_J \hat{q} \sim 1 \ , \ \partial_J q \sim 0$ and

$$\eta\partial_J\hat{v} \sim (\hat{v}\partial_J\hat{q} + \hat{q}\partial_J\hat{v}) \operatorname{tr}[\hat{v}\boldsymbol{I} + \hat{q}\boldsymbol{\Sigma}_M]^{-1}\boldsymbol{\Sigma}_M - \hat{q}\hat{v}\operatorname{tr}[\hat{v}\boldsymbol{I} + \hat{q}\boldsymbol{\Sigma}]^{-2}[\partial_J\hat{v}\boldsymbol{\Sigma}_M + \partial_J\hat{q}\boldsymbol{\Sigma}_M^2]$$
$$\sim \hat{q}^2\operatorname{tr}[\hat{v}\boldsymbol{I} + \hat{q}\boldsymbol{\Sigma}_M]^{-2}\boldsymbol{\Sigma}_M^2\partial_J\hat{v} + \hat{v}^2\operatorname{tr}[\hat{v}\boldsymbol{I} + \hat{q}\boldsymbol{\Sigma}_M]^{-2}\boldsymbol{\Sigma}_M$$
$$\Rightarrow \partial_J\hat{v} \sim \frac{\operatorname{tr}[\boldsymbol{I} + a^{-1}\lambda^{-1}\boldsymbol{\Sigma}_M]^{-2}\boldsymbol{\Sigma}_M}{\eta - \operatorname{tr}[a\lambda\boldsymbol{I} + \boldsymbol{\Sigma}_M]^{-2}\boldsymbol{\Sigma}_M^2}. \tag{68}$$

We note that $\partial_J\hat{v}$ only depends on the product $a\lambda$ which is an implicit function of $\eta$ and $\boldsymbol{\Sigma}_M$. The generalization error is $E_g = \frac{1}{M}\partial_J\hat{q}(1 + \hat{v})\boldsymbol{w}^*[(1 + \hat{v})\boldsymbol{I} + \hat{q}\boldsymbol{\Sigma}]^{-1}\boldsymbol{\Sigma}_M\boldsymbol{w}^*$

$$E_g \sim \frac{1}{M}\boldsymbol{w}^* \left[\boldsymbol{I} + a^{-1}\lambda^{-1}\boldsymbol{\Sigma}_M\right]^{-2}\boldsymbol{\Sigma}_M\boldsymbol{w}^*$$
$$+ \frac{1}{M}\boldsymbol{w}^*[\lambda a\boldsymbol{I} + \boldsymbol{\Sigma}_M]^{-2}\boldsymbol{\Sigma}_M^2\boldsymbol{w}^* \times \frac{\operatorname{tr}[\boldsymbol{I} + a^{-1}\lambda^{-1}\boldsymbol{\Sigma}_M]^{-2}\boldsymbol{\Sigma}_M}{\eta - \operatorname{tr}[a\lambda\boldsymbol{I} + \boldsymbol{\Sigma}_M]^{-2}\boldsymbol{\Sigma}_M^2}. \tag{69}$$

We see that in the generic case, the asymptotic error has a nontrivial dependence on the task $\boldsymbol{w}^*$ and the correlation structure $\boldsymbol{\Sigma}_M$. To gain more intuition, we will now consider the special case of isotropic features $\boldsymbol{\Sigma}_M = \boldsymbol{I}$. In this case, we have $\eta = \frac{1}{1+\lambda a}$ so that $\lambda a = \frac{1-\eta}{\eta}$. This results in the following generalization error

$$E_g \sim \frac{1}{M}|\boldsymbol{w}^*|^2 \left[(1 - \eta)^2 + \eta^2\frac{(1 - \eta)^2}{\eta - \eta^2}\right] \sim \frac{1}{M}|\boldsymbol{w}^*|^2(1 - \eta). \tag{70}$$

We see that as $\eta = \frac{N_\mathcal{H}}{M} \to 1$, the asymptotic error converges to zero since all information in the original features is preserved.

### D.5.2 Simplified Isotropic Feature Noise

We can simplify the above expressions somewhat in the case where $\sigma^2 = 1$ and $\boldsymbol{\Sigma}_\epsilon = \sigma_\epsilon^2\boldsymbol{I}$. In this case, the order parameters become

$$\eta v = \eta(1 + \hat{v} + \sigma_\epsilon^2\hat{q})^{-1} \implies v = \frac{1}{1 + \hat{v} + \sigma_\epsilon^2\hat{q}}$$

$$\implies \eta\hat{v} = \hat{q} \operatorname{tr}\left[\boldsymbol{I} + \frac{\hat{q}}{1 + \hat{v} + \sigma_\epsilon^2\hat{q}}\boldsymbol{\Sigma}_M\right]^{-1}\boldsymbol{\Sigma}_M = \hat{q}(1 + \hat{v} + \sigma_\epsilon^2\hat{q}) \operatorname{tr}[(1 + \hat{v} + \sigma_\epsilon^2\hat{q})\boldsymbol{I} + \hat{q}\boldsymbol{\Sigma}_M]^{-1}\boldsymbol{\Sigma}_M$$

$$q = \operatorname{tr}\left[(1 + \hat{v} + \sigma_\epsilon^2\hat{q})\boldsymbol{I} + \hat{q}\boldsymbol{\Sigma}_M\right]^{-1}\boldsymbol{\Sigma}_M + \frac{\eta\sigma_\epsilon^2}{1 + \hat{v} + \sigma_\epsilon^2\hat{q}}. \tag{71}$$

Letting $\boldsymbol{G} = [(1 + \hat{v} + \sigma_\epsilon^2 \hat{q})\boldsymbol{I} + \hat{q}\boldsymbol{\Sigma}_M]^{-1}$, the source derivatives have the form

$$\partial_J \hat{q} = 1 - \frac{\alpha}{(\lambda + q)^2} \partial_J q \tag{72}$$

$$= 1 + \frac{\alpha}{(\lambda + q)^2} \left[ \text{tr}\boldsymbol{G}^2\boldsymbol{\Sigma}[\partial_J \hat{v}\boldsymbol{I} + \partial_J \hat{q}\boldsymbol{\Sigma}_M] + \frac{\eta\sigma_\epsilon^2}{(1 + \hat{v} + \sigma_\epsilon^2\hat{q})^2}(\partial_J \hat{v} + \sigma_\epsilon^2 \partial_J \hat{q}) \right],$$

$$\eta\partial_J \hat{v} = ((1 + \hat{v} + 2\sigma_\epsilon^2\hat{q})\partial_J\hat{q} + \hat{q}\partial_J\hat{v})\text{tr}\boldsymbol{G}\boldsymbol{\Sigma}_M \tag{73}$$

$$- \hat{q}(1 + \hat{v} + \sigma_\epsilon^2\hat{q})\text{tr}\boldsymbol{G}^2\boldsymbol{\Sigma}_M[(\partial_J\hat{v} + \sigma_\epsilon^2\partial_J\hat{q})\boldsymbol{I} + \partial_J\hat{q}\boldsymbol{\Sigma}_M]$$

$$= (\partial_J\hat{q})(1 + \hat{v} + \sigma_\epsilon^2\hat{q})^2\text{tr}\boldsymbol{G}^2\boldsymbol{\Sigma} + (\partial_J\hat{v} + \sigma_\epsilon^2\partial_J\hat{q})\hat{q}^2\text{tr}\boldsymbol{G}^2\boldsymbol{\Sigma}^2. \tag{74}$$

This is a $2 \times 2$ linear system

$$\begin{bmatrix} 1 - \frac{\alpha}{(\lambda+q)^2}[\text{tr}\boldsymbol{G}^2\boldsymbol{\Sigma}^2 + \frac{\eta\sigma_\epsilon^4}{(1+\hat{v}+\sigma_\epsilon^2\hat{q})^2}] & -\frac{\alpha}{(\lambda+q)^2}[\text{tr}\boldsymbol{G}^2\boldsymbol{\Sigma} + \frac{\eta\sigma_\epsilon^2}{(1+\hat{v}+\sigma_\epsilon^2\hat{q})^2}] \\ -(1 + \hat{v} + \sigma_\epsilon^2\hat{q})^2\text{tr}\boldsymbol{G}^2\boldsymbol{\Sigma}_M - \sigma_\epsilon^2\hat{q}^2\text{tr}\boldsymbol{G}^2\boldsymbol{\Sigma}_M^2 & \eta - \hat{q}^2\text{tr}\boldsymbol{G}^2\boldsymbol{\Sigma}_s^2 \end{bmatrix} \begin{bmatrix} \partial_J\hat{q} \\ \partial_J\hat{v} \end{bmatrix} = \begin{bmatrix} 1 \\ 0 \end{bmatrix}.$$

For each $\alpha$, we can solve for $\partial_J\hat{q}$ and $\partial_J\hat{v}$ to get the final generalization error with the formula

$$E_g = \partial_J \frac{1}{M}\boldsymbol{w}^* \left[ (1 + \hat{v} + \sigma_\epsilon^2\hat{q})\hat{q}\boldsymbol{\Sigma}\boldsymbol{G} \right] \boldsymbol{w}^*$$

$$= \frac{1}{M}\boldsymbol{w}^*[\partial_J(\hat{q} + \hat{q}\hat{v} + \sigma_\epsilon^2\hat{q}^2)\boldsymbol{\Sigma}\boldsymbol{G} - (1 + \hat{v} + \sigma_\epsilon^2\hat{q})\hat{q}\boldsymbol{\Sigma}\boldsymbol{G}^2(\partial\hat{v}\boldsymbol{I} + \sigma_\epsilon^2\partial\hat{q}\boldsymbol{I} + \partial\hat{q}\boldsymbol{\Sigma}_M)]\boldsymbol{w}^*. \tag{75}$$

An example of these solutions can be found in Figure 16, where we show good agreement between theory and experiment.

# E  RESNET ON CIFAR EXPERIMENTS

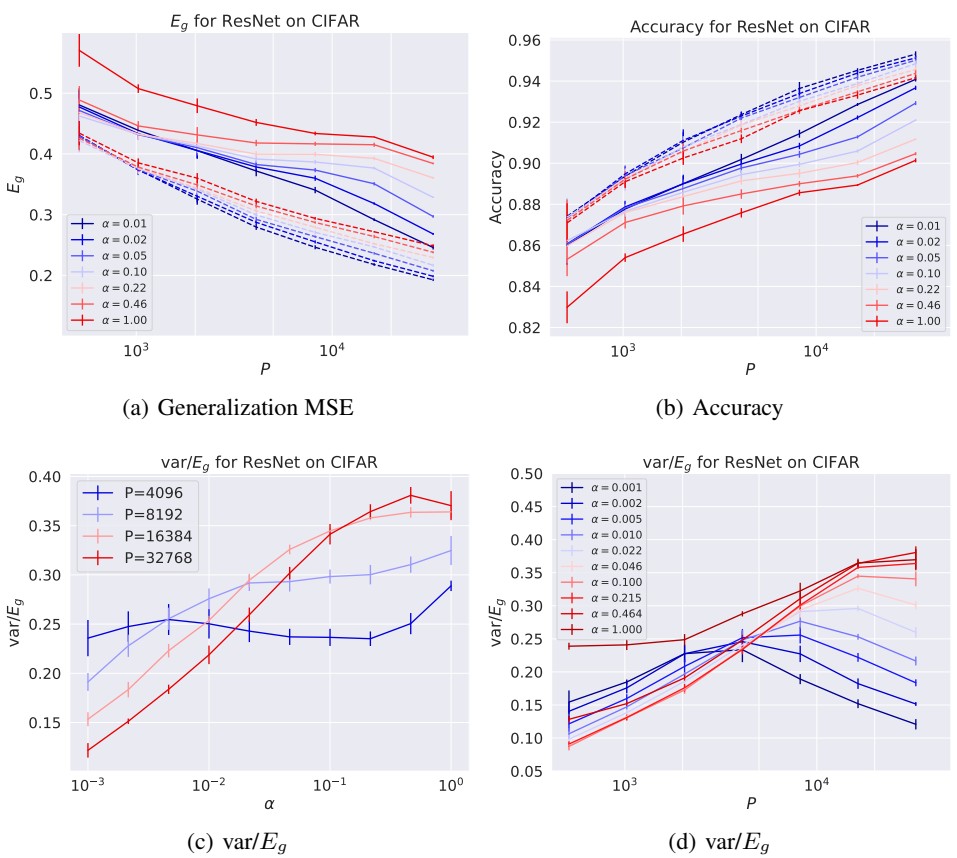

Figure 8: A Wide ResNet Zagoruyko & Komodakis (2017) trained on a superclassed CIFAR task comparing animate vs inanimate objects. Each learning curve is averaged over 5 different samples of the train set, yielding the means and error bars shown in the figures. a) Generalization error $E_g$. The dashed lines are the error of a 20-fold ensemble over different values of $\alpha$. Across all $P$, lazy networks attain worse generalization error. As with the MLP task, the best performing networks are ensembles of rich networks. b) The accuracy also has the same trend: richer networks perform better and ensembling lazy networks helps them more. c) Once $P$ is large enough, lazier networks tend to benefit more from ensembling. d) Very lazy networks transition to variance limited behavior earlier. For ResNets on this task, we see that rich, feature learning networks eventually begin reducing their variance on this task. Further details of the experiment are given in section A.1.

# F  ADDITIONAL EXPERIMENTS

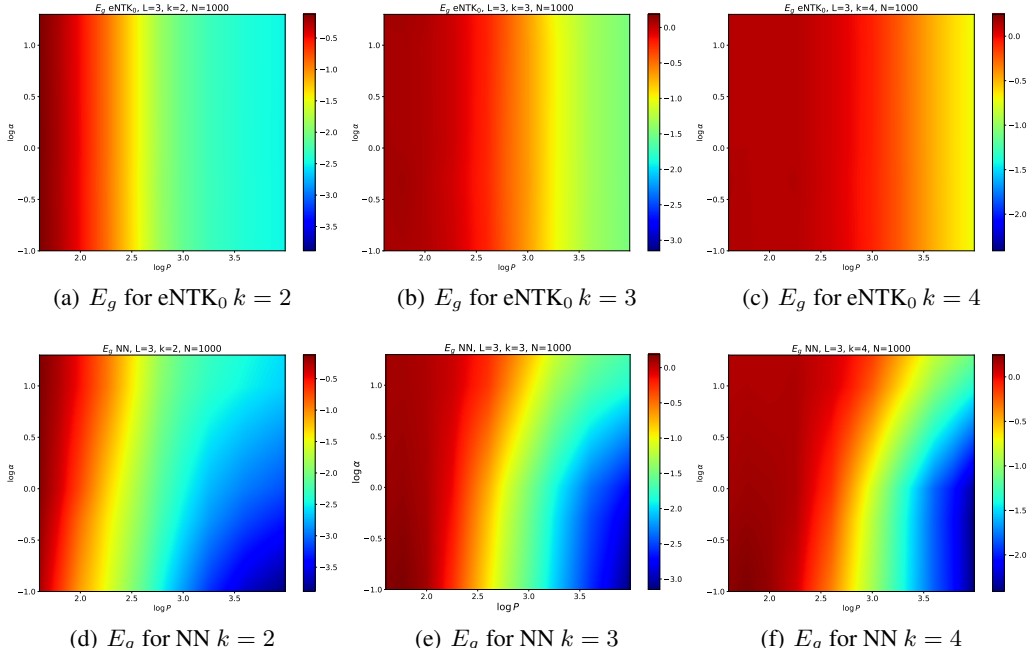

Figure 9: Phase plots of $\log_{10} E_g$ for initial eNTKs (top) and neural networks (bottom). The large $\alpha$ behavior of the neural network generalization matches the generalization of the corresponding eNTK$_0$. As a sanity check, the eNTK$_0$ generalization error is independent of re-scaling the network initialization because of the homogeneity of the ReLU network output.

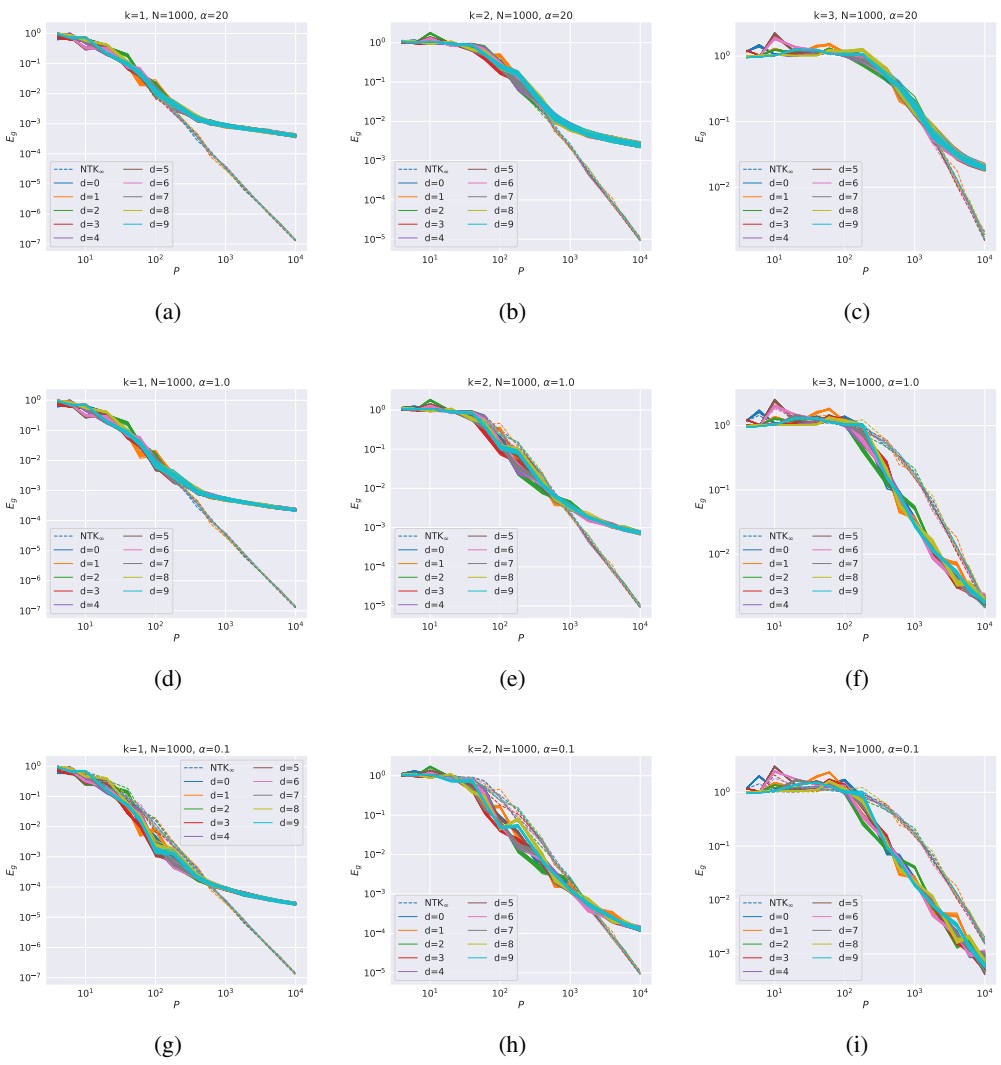

Figure 10: A fine-grained view of the generalization error across different datasets and ensembles. Solid curves are depth 3 neural networks, dashed curves are the infinite width NTK (which only has variance over datasets). Each color is a set of networks trained on the same dataset but different initializations. Different colors correspond to different datasets indexed by $d \in \{0, \ldots, 9\}$.

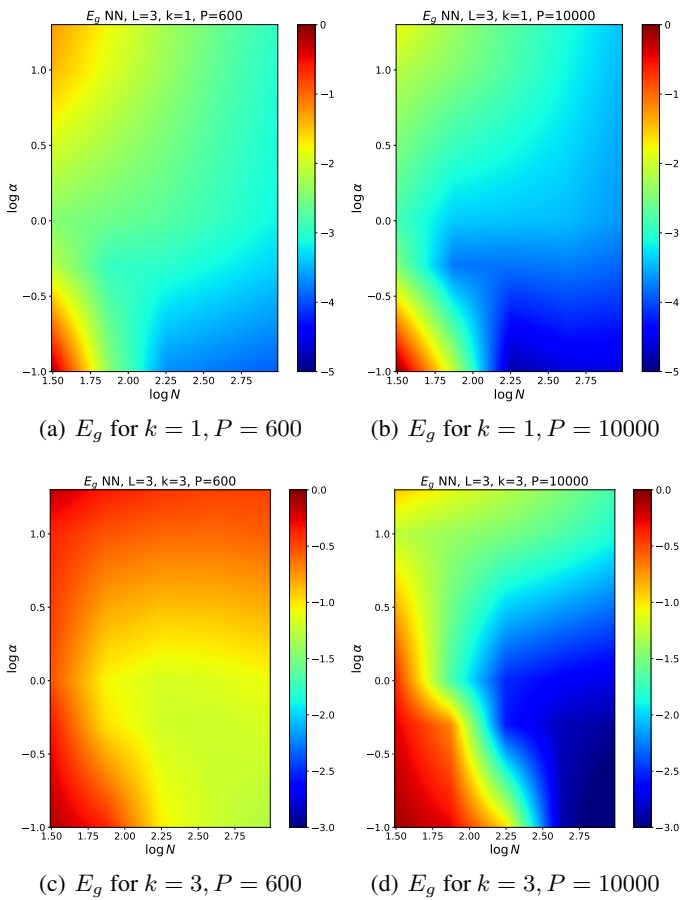

(a) $E_g$ for $k = 1, P = 600$    (b) $E_g$ for $k = 1, P = 10000$

(c) $E_g$ for $k = 3, P = 600$    (d) $E_g$ for $k = 3, P = 10000$

Figure 11: Phase plots of $\log_{10} E_g$ for neural networks in the $N$-$\alpha$ plane. We plot these at different train set sizes $P$ and different tasks $k$. The colors are fixed to match across networks trained on the same task.

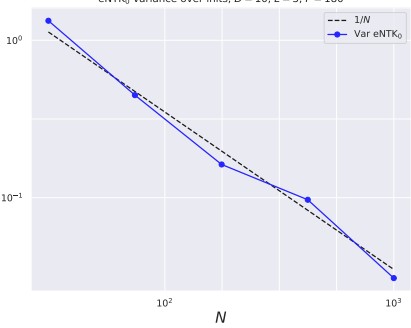

Figure 12: Empirical plot of of the scaling of the variance of the $eNTK_0$ with $N$ with variance taken over 10 initializations and averaged 10 different datasets.

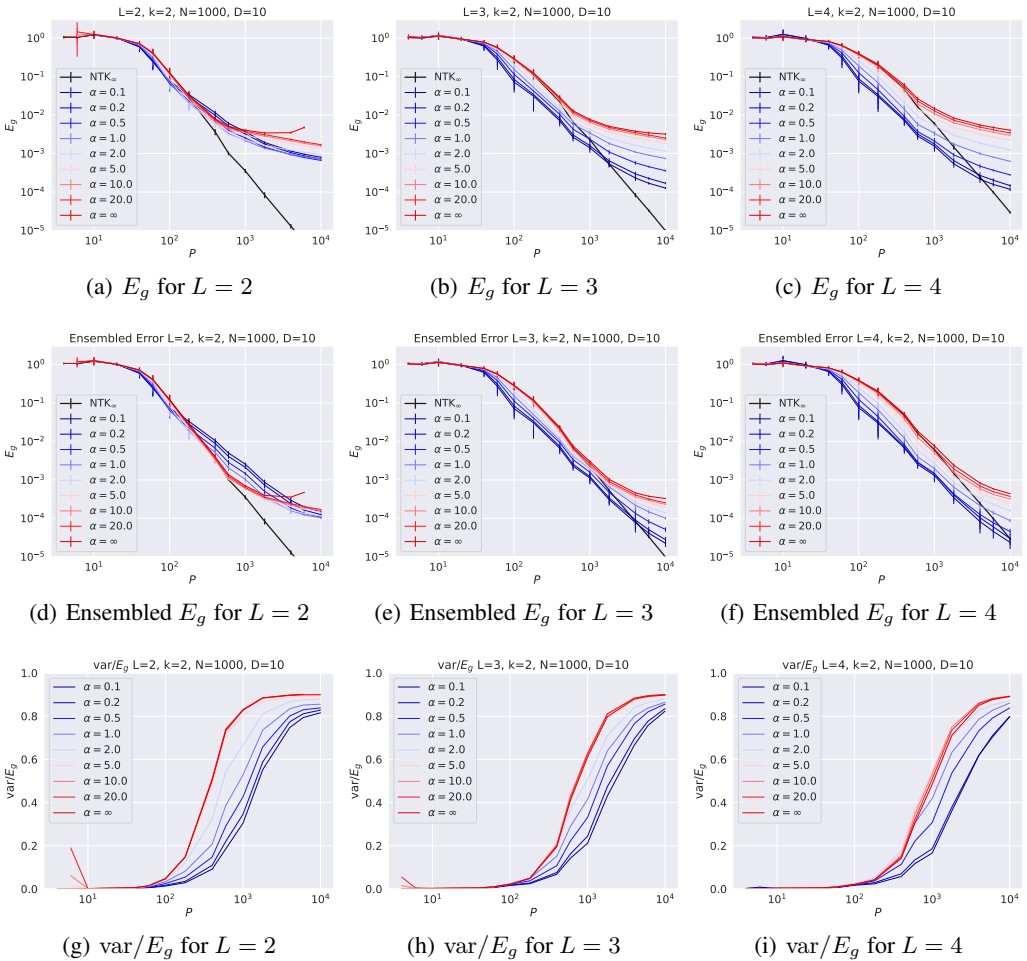

(a) $E_g$ for $L = 2$      (b) $E_g$ for $L = 3$      (c) $E_g$ for $L = 4$

(d) Ensembled $E_g$ for $L = 2$      (e) Ensembled $E_g$ for $L = 3$      (f) Ensembled $E_g$ for $L = 4$

(g) var/$E_g$ for $L = 2$      (h) var/$E_g$ for $L = 3$      (i) var/$E_g$ for $L = 4$

Figure 13: Sweep over depth $L = \{2, 3, 4\}$. Deeper networks in the rich regime can more easily outperform the infinite width network for a larger range of $P$. Also, for larger $L$ it is easier to deviate from the lazy regime at a given $\alpha$. By contrast, on this task the shallower NTK$_\infty$ outperforms deeper NTK$_\infty$s. As before, ensembled lazy networks approach NTK$_\infty$ and the variance rises with $P$.

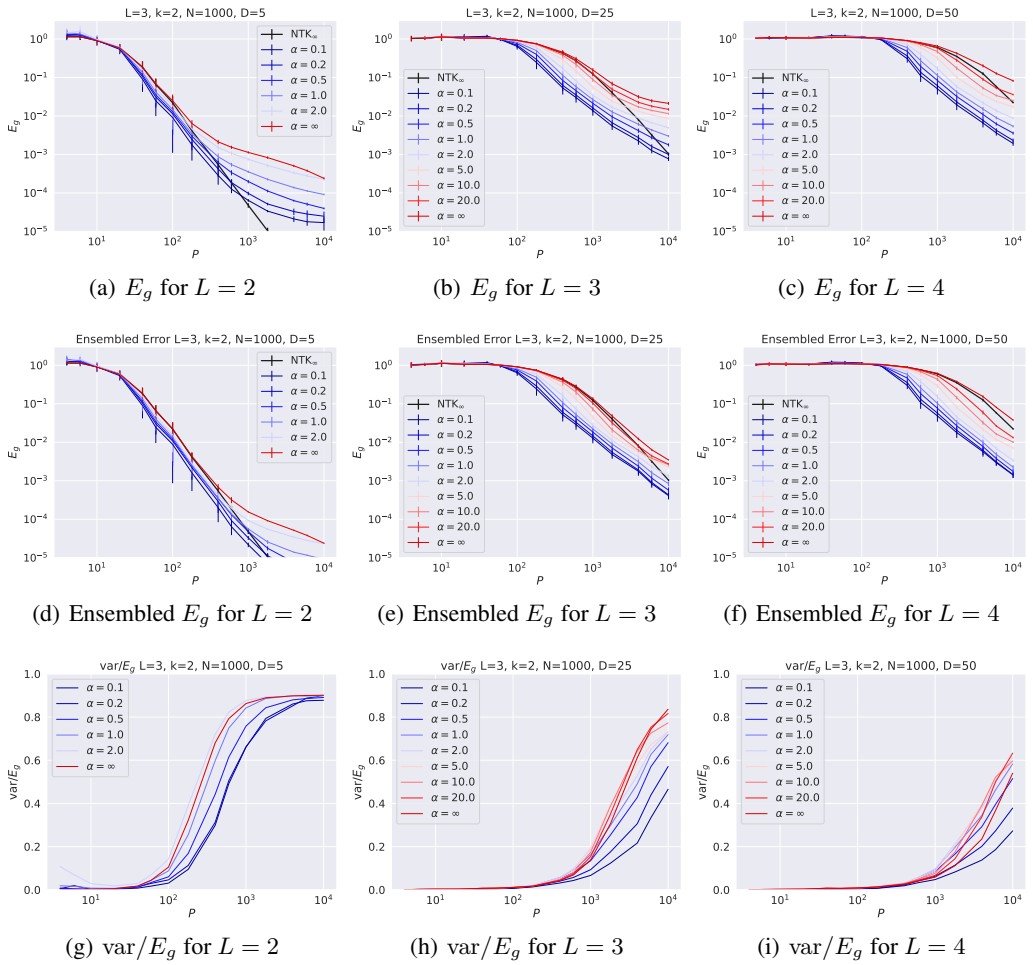

Figure 14: Sweep over input dimension $D = \{5, 25, 50\}$. At larger input dimensions rich networks can more easily outperform $\text{NTK}_\infty$. This is a consequence of the task depending on the low-dimensional projection $\boldsymbol{\beta} \cdot \boldsymbol{x}$.

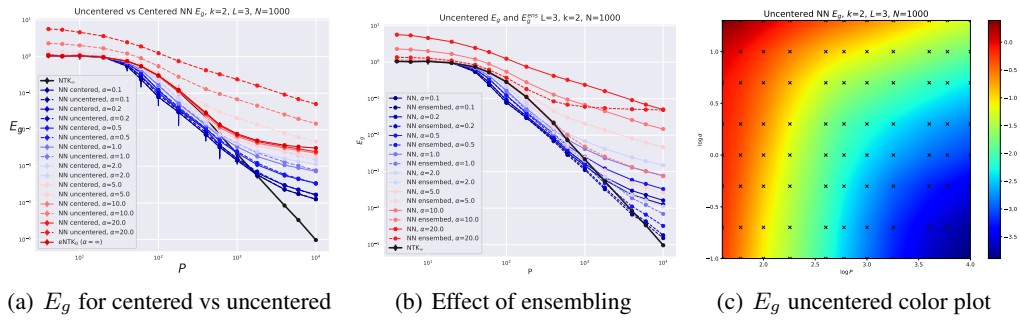

(a) $E_g$ for centered vs uncentered    (b) Effect of ensembling    (c) $E_g$ uncentered color plot

Figure 15: a) $E_g$ for the centered predictor $\tilde{f}_\theta(\boldsymbol{x}) - \tilde{f}_{\theta_0}(\boldsymbol{x})$ (solid) compared to the generalization of the uncentered predictor $\tilde{f}_\theta(\boldsymbol{x})$ (dashed). At small $\alpha$, the difference is negligible, while at large $\alpha$ the uncentered predictor does worse and does not approach $\text{eNTK}_0$. The worse generalization can be understood as $\tilde{f}_{\theta_0}(\boldsymbol{x})$ effectively adding an initialization-dependent noise to the target $\boldsymbol{y}$. b) The effect of ensembling becomes less beneficial for uncentered lazy networks. c) Color plot of $E_g$. The lazy regime is different from the $\text{eNTK}_0$ generalization (c.f. Figure 9).

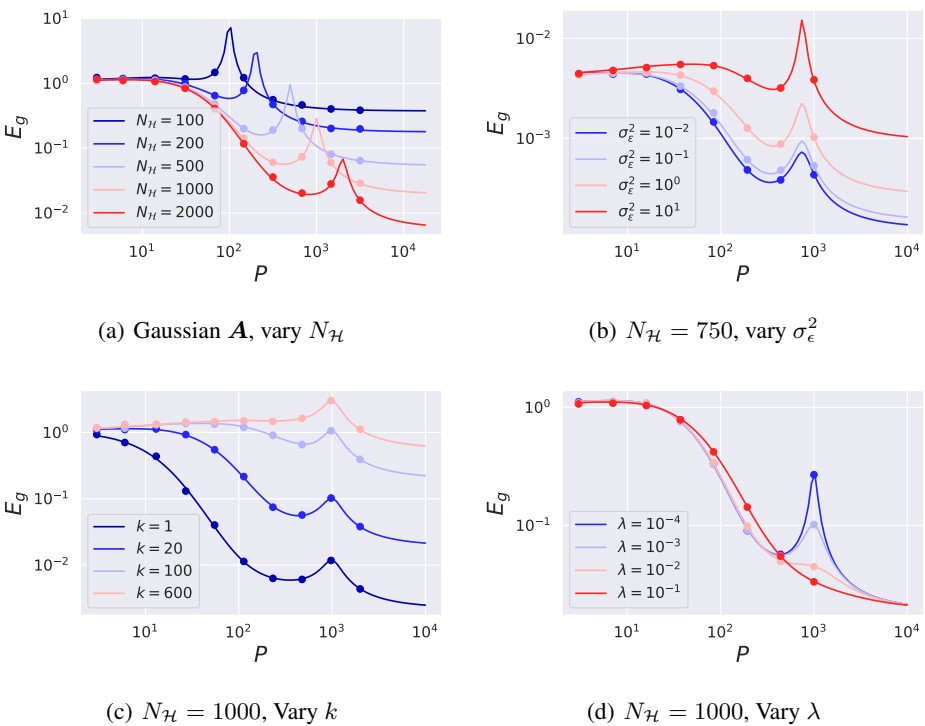

(a) Gaussian $A$, vary $N_{\mathcal{H}}$

(b) $N_{\mathcal{H}} = 750$, vary $\sigma_\epsilon^2$

(c) $N_{\mathcal{H}} = 1000$, Vary $k$

(d) $N_{\mathcal{H}} = 1000$, Vary $\lambda$

Figure 16: Verification of Gaussian $A$ model. Solid lines are theory and dots are experiments. (a) The effect of changing the student's RKHS dimension $N_{\mathcal{H}}$. Double descent overfitting peaks occur at $P = N_{\mathcal{H}}$ (b) The effect of additive noise in the student features $\Sigma_\epsilon = \sigma_\epsilon^2 \Sigma_M$. (c) Learning curves for fitting the $k$-th eigenfunction. All mode errors exhibit a double descent peak at $P = N_{\mathcal{H}}$ regardless of the task. (d) Regularization can prevent the overfitting peak.

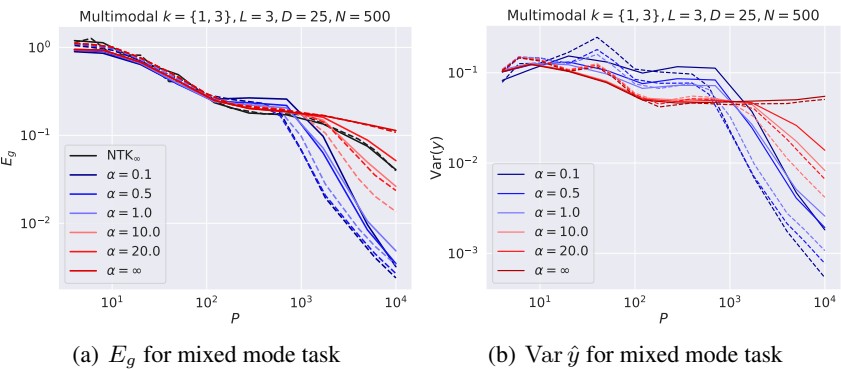

(a) $E_g$ for mixed mode task

(b) Var $\hat{y}$ for mixed mode task

Figure 17: Width 500 depth 3 MLP learning a $D = 25$ mixture of a linear and cubic polynomials. a) Generalization error of $\text{NTK}_\infty$ (solid black) and MLP (solid colored lines) on mixed mode task. The dashed lines are convex combinations of the generalization curves for the pure mode $k = 1, k = 3$ tasks. For the $\text{NTK}_\infty$, the generalization curves sum to give the mixed mode curve, as observed in Bordelon et al. (2020). We see that this also holds for the $\text{eNTK}_0$ for sufficiently lazy networks, as predicted by the simple renadom feature model considred in section 4 of this paper. b) The variance curves for the same task. Again, for sufficiently lazy networks the variance is a sum of the variances of the individual pure mode tasks, as predicted by our random feature model.

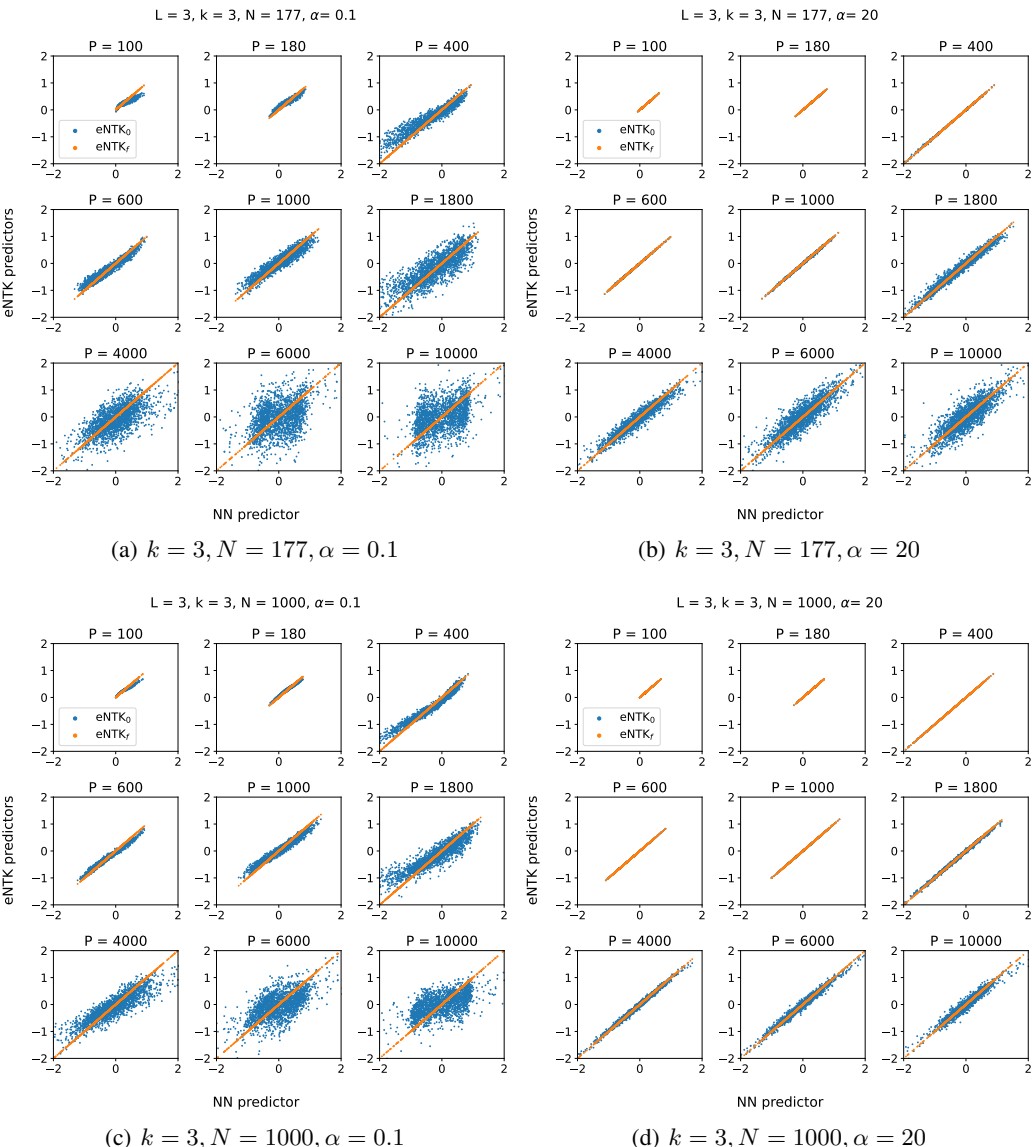

Figure 18: Comparison of the neural network predictor ($x$-axis) to eNTK$_0$ , eNTK$_f$ (blue, orange respectively, $y$-axis) across several training dataset sizes. a) $N = 177, \alpha = 0.1$, b) $N = 177, \alpha = 20$, c) $N = 1000, \alpha = 0.1$ d) $N = 1000, \alpha = 20$. All networks are depth 3. Note how in the rich regime there is a much stronger distinction between the eNTK$_0$ and the neural network. In all regimes, eNTK$_f$ matches the NN output.

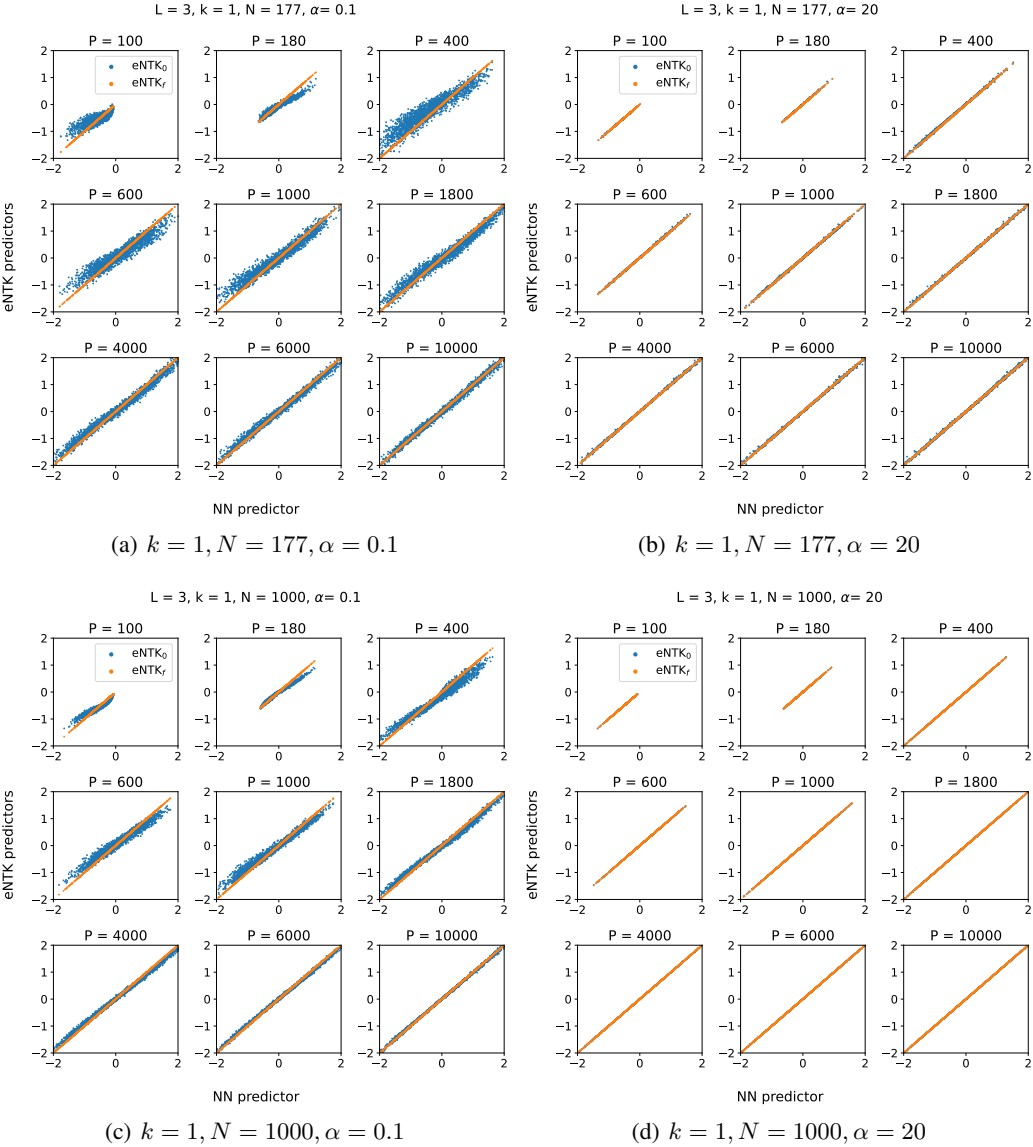

Figure 19: The same as figure 18 but for fitting a linear $k = 1$ mode. Because the task is simpler, it doesn't require as large of an $\alpha$ to enter the lazy regime.

