# OpenReview forum: "The Onset of Variance-Limited Behavior for Networks in the Lazy and Rich Regimes"
_ICLR.cc/2023/Conference — ICLR 2023 poster_

### Official Review · Reviewer_JK78 · 2022-10-20

**Confidence:** 4
**Correctness:** 4
**Technical Novelty And Significance:** 3
**Empirical Novelty And Significance:** 3
**Recommendation:** 8

**Clarity, Quality, Novelty And Reproducibility:**

The article is well written and the figures are clear.

The paper references the line of works that study the variance for finite widths (especially in the lazy regime), as well as feature learning in the rich regime. The novelty of this article mostly lies in the study of these effects accross regimes to give a more complete overview.

**Strength And Weaknesses:**

With the recent focus on understanding infinite width networks, it is crucial to understand their discrepancy with finite width networks. The paper makes a number of interesting observation, supported by detailed experiments and a small toy model.

While the authors have quite thoroughly studied the impact the of the number of datapoints $P$, width $N$ and initialization, the effect of the depth and of the task itself (input dimension, degree of the polynomial or more general function) are not really studied. Of course, I understand that it is not possible to study all of these effect simultaneously in a 9 page paper.

**Summary Of The Paper:**

The paper compares the generalization of finite width networks to their infinite width (NTK) counterpart. More precisely the paper analyse empirically the effect of increasing the number of datapoints $P$ for some fixed networks. These finite width effects depend on the regime (lazy or rich) the network are in:
 - In the lazy regime, the difference between finite width and infinite width network is due to an increase in variance for low widths. As a result the performence of finite width networks is strictly worse than the NTK limit.
 - In the rich regime, the presence of feature-learning (which is absent in the infinite-width NTK limit) helps generalization, thus outperforming the NTK limit for a range of dataset sizes, but for large $P$ the variance dominates, thus leading to worse performence.
 - The authors observe that the start of this variance dominated regime scales roughly as $P=\sqrt N$ in both regimes, though it starts later in the rich regime.

The authors present a toy model to mathematically recover some of the properties observed empirically.

**Summary Of The Review:**

The paper studies finite width effects in DNNs, in particular the opposing effects of feature learning (which improves generalization) and of the variance (which worsens generalization). These two effects remain ill understood in deep neural networks (though many paper have studied them) and their interaction is even less understoood. The paper study both of these effects empirically and with a few small theoretical results.

---

> ### Author Response · Authors · 2022-11-17
> **Response to Reviewer JK78**
>
> *While the authors have quite thoroughly studied the impact the of the number of datapoints , width  and initialization, the effect of the depth and of the task itself (input dimension, degree of the polynomial or more general function) are not really studied. Of course, I understand that it is not possible to study all of these effect simultaneously in a 9 page paper.*
>
> We thank the reviewer for the supportive review as well as for these helpful comments and suggestions. While it is indeed difficult to sweep over all parameters simultaneously, we have incorporated additional sweeps over input dimension $D$ and depth $L$ for $k=2$ tasks in Appendix Figures 13 and 14. There, we have plotted the learning curves and demonstrated trends consistent with those in the main text. In those figures, we note that feature learning provides greater advantages in high dimensions. Further, we note that deeper networks also have larger differences in generalization error between feature learning and lazy networks. The general observations about the variance contributions continue to hold.
>
> To vary the task, we have also added an experiment where mixtures of polynomials of different degrees are learned. We see the same general trends with $\alpha, P$. The learning curves and variance plots are provided in Figure 16. Finally, we have added learning curves of a ResNet trained on CIFAR in Figure 8.

---

### Official Review · Reviewer_4Na8 · 2022-10-22

**Confidence:** 2
**Correctness:** 4
**Technical Novelty And Significance:** 3
**Empirical Novelty And Significance:** 3
**Recommendation:** 8

**Clarity, Quality, Novelty And Reproducibility:**

**Clarity**

The work is clear and well written.

**Quality**

The experiments are of high quality.

**Originality**

The experiments are well placed relative to the literature as far as I can tell.

**Reproducibility**

The work states that "We plan to make the results of our
experiments publicly accessible, alongside the code to generate them."  If this is followed up upon this will support reproducibility.

**Questions for the authors:**

The authors claim on page 8 “First, we note that the noise correlations $\Sigma_n$ should scale as $O(1/N)$ since kernels, either in the lazy or rich regime, exhibit variance $1/N$ fluctuations over initialization”.  Perhaps I am unaware of more recent results that provide stronger bounds but as far as I know the deviations of the NTK at initialization are $O(1/\sqrt{N})$, see e.g. [2], [3]. [4], [5].  The claim that kernel deviations are $O(1/N)$ is made multiple times in the paper, for example on page 6 “In our case, the kernel variance at initialization scales as $\sigma^{2L}/N$.  Could you please provide a reference or at least make this claim precise?

References:

[1] Lee et al.  Finite Versus Infinite Neural Networks: an Empirical Study.

[2] Simon S. Du, Xiyu Zhai, Barnabas Poczos, and Aarti Singh. Gradient descent provably optimizes over-parameterized neural networks. In International Conference on Learning Representations, 2019.

[3] Simon Du, Jason Lee, Haochuan Li, Liwei Wang, and Xiyu Zhai. Gradient descent finds global minima of deep neural networks.

[4] Jiaoyang Huang and Horng-Tzer Yau. Dynamics of deep neural networks and neural tangent hierarchy.

[5] Sam Buchanan, Dar Gilboa, and John Wright. Deep networks and the multiple manifold problem.



**Strength And Weaknesses:**

**Strengths:**

This work considers the learning curves for finite widths and finite sample sizes, in contrast to previous works that require that one of the quantities tends to infinity.
There are plenty of experimental results with clear figures and illustrations.
The exposition is clear

**Weaknesses:**

The structure of the data (degree $k$ polynomials on the sphere), uniformly distributed inputs is a bit limited.  In Figure 1 we already see that increasing the degree of the polynomial can already change the results for the ensembled networks (for $k=2$ the ensembles performs like the infinite-width kernel, for $k = 4$ the ensembles outperform).  It would be reassuring if we had an idea of which results are robust to the data distribution and target function.
The results may be specific to feedforward networks.  As was shown in [1] gap between NTK performance and finite width networks is greater for CNNs in comparison with feedforward networks.  While ensembled feedforward networks perform like the NTK, ensembled convolutional nets significantly outperform the NTK.
I do have one reservation about the variance of the NTK over initialization.  Please see my question about the NTK variance in the questions section.


**Summary Of The Paper:**

This paper studies the generalization error of deep ReLU feedforward networks trained with different initialization scales, widths, and training set sizes in comparison to kernel regression with Neural Tangent Kernels (the kernel in the infinite-width limit, the kernel at initialization, and the kernel at the end of training).  They demonstrate that the kernel corresponding to the infinite-width limit outperforms lazy networks (large initialization scales) and the kernel at initialization.  Furthermore finite width networks even in the rich regime (small initialization scales) are outperformed by the infinite-width NTK at large dataset sizes, however rich networks can outperform at intermediate dataset sizes.  Furthermore they study the dataset size after which most of the generalization error comes from the variance arising from the parameter initialization, and conclude that the transition occurs at dataset sizes on the order of the square root of the network width.  Consistent with other works, they show that the NTK at the end of training exhibits similar generalization error to the trained network (for a variety of widths and initialization scales).

**Summary Of The Review:**

**Initial Recommendation:**

Accept.  While the data model is a bit limited, the paper makes an interesting study of comparing networks in the lazy and rich regimes to NTK methods at different dataset sizes.

**Comments/Additional feedback:**

Figure 2 (a) On the top of the figure worth replacing $E_g$ with $\log(E_g)$ since it is the log generalization error

Figure 3 (b) What width were the various networks trained with for the different $\alpha$ values.  Was the width $N$ fixed and alpha varied to plot the stars?

Page 7 Perhaps use \eqref to reference equation 3

---

> ### Author Response · Authors · 2022-11-17
> **Response to Reviewer 4Na8**
>
> ### Strengths:
>
> *This work considers the learning curves for finite widths and finite sample sizes, in contrast to previous works that require that one of the quantities tends to infinity. There are plenty of experimental results with clear figures and illustrations. The exposition is clear*
>
> ### Weaknesses:
>
> *The structure of the data (degree $k$ polynomials on the sphere), uniformly distributed inputs is a bit limited. In Figure 1 we already see that increasing the degree of the polynomial can already change the results for the ensembled networks (for $k=2$ the ensembles performs like the infinite-width kernel, for $k=4$ the ensembles outperform). It would be reassuring if we had an idea of which results are robust to the data distribution and target function. The results may be specific to feedforward networks. As was shown in [1] gap between NTK performance and finite width networks is greater for CNNs in comparison with feedforward networks. While ensembled feedforward networks perform like the NTK, ensembled convolutional nets significantly outperform the NTK. I do have one reservation about the variance of the NTK over initialization. Please see my question about the NTK variance in the questions section.*
>
> We thank the reviewer for this feedback. We agree that polynomial regression on the hypersphere is not fully realistic for a deep learning problem. To extend our framework and experiments to more realistic settings, we have added ensembling experiments for Wide ResNets on CIFAR-10 in Figure 8 of the appendix.  In the new figure, we plot the test loss as function of $\alpha$ and $P$, for width (channel count) $N = 64$ CNNs. Consistent with the rest of our paper, we find that richer networks generalize better and that feature learning provides a reduction in variance due to initialization.  We see that ensembling gives a reduction in test error that is more prominent for lazy networks.
>
> ### Questions for the authors:
>
> *The authors claim on page 8 “First, we note that the noise correlations  $\Sigma_n$ should scale as $O(1/N)$ since kernels, either in the lazy or rich regime, exhibit variance $1/N$ fluctuations over initialization”. Perhaps I am unaware of more recent results that provide stronger bounds but as far as I know the deviations of the NTK at initialization are $O(1/\sqrt N$, see e.g. [2], [3]. [4], [5]. The claim that kernel deviations are $O(1/N)$ is made multiple times in the paper, for example on page 6 “In our case, the kernel variance at initialization scales as $\sigma^{2L}/N$. Could you please provide a reference or at least make this claim precise?*
>
> We thank the reviewer for pointing out our imprecise use of language on this point. We intended to claim throughout the work that the *variance* of kernel entries are $O(1/N)$ but that the *standard deviations* or scale of fluctuations are $O(N^{-1/2})$ as the reviewer indicates. This would correspond to noise correlation matrix which has scale $1/N$, since this is a measure of variance.
>
> Throughout the paper, we have corrected any reference to the scale of the kernel fluctuations and made clear that we mean *variance* $O(1/N)$. Additionally, in Figure 12 we empirically verify that this variance scaling holds for the initial eNTK as a function of $N$.
>
> ### Comments/Additional feedback:
>
> 1. *Figure 2 (a) On the top of the figure worth replacing $E_g$ with $\log(E_g)$ since it is the log generalization error*
>
> Thanks for this catch. We have fixed the title to appropriately reflect the log scale of the color plot.
>
> 2. *Figure 3 (b) What width were the various networks trained with for the different  values. Was the width  fixed and alpha varied to plot the stars?*
>
> We thank the reviewer for pointing out the confusing labeling of the markers in that figure. We have added text to the caption to highlight that the different colors correspond to different $\alpha$ while the different marker types correspond to different widths.
>
> 3. *Page 7 Perhaps use \eqref to reference equation 3*
>
> We thank the reviewer for this catch. We have resolved this issue. The [3] referred to a figure earlier in the text. The sentence has been made clearer.

---

> > ### Comment · Reviewer_4Na8 · 2022-11-17
> > **Response to authors**
> >
> > I would like to thank the authors for their comments and clarifications.  I have no further questions at this time.  Good work.

---

### Official Review · Reviewer_6LDR · 2022-10-30

**Confidence:** 2
**Correctness:** 4
**Technical Novelty And Significance:** 2
**Empirical Novelty And Significance:** 2
**Recommendation:** 6

**Clarity, Quality, Novelty And Reproducibility:**

**Questions:**
- Abstract. "However, at a critical sample size P∗, the finite-width network generalization begins to worsen compared to the infinite width performance." Is it that the generalization necessarily worsens, or that the generalization bounds that have been proven are worse than the generalization bounds for the corresponding infinite width model?
- "we demonstrate empirically that all networks have the same generalization error as kernel regression solutions with their final eNTKs. A". It is not obvious to me why networks would \emph{not} have the same generalisation error as kernel regression using final eNTKs. Can you maybe add a sentence explaining why this is the case? Perhaps in general one would expect the kernel regression to be closer to the GPNN kernel (in the sense of Neal, Matthews, Lee, etc.) rather than the NTK (Jacot etc.)?
- Around equation 2. "There we also show that the O(1/N) term is strictly positive for sufficiently large N. Thus, for lazy networks, finite width effects lead to strictly worse generalization error." Wouldn't it be more appropriate to say that "for lazy networks \emph{of sufficiently large $N$}, finite widht effects lead to strictly worse generalization error." Since the error term is strictly positive only for sufficiently wide networks, less wide networks might have better generalisation error.


**Minor:**
- Last sentence of conclusion. "and number of ensembles of a network can be tuned to achieve optimal performance under a fixed compute and data budget." Should this say something more like "number of networks in an ensemble"?
- Somewhere at the beginning, you introduce the terminology "final empirical NTK" with a symbol like eNTK_f. Instead of repeating "final empirical NTK", you could just write the symbol eNTK_f. Alternatively, repeat the terminology. At the moment, you use both (eg. the next dot point after you introduce the terminology). If you repeat both, it is not as clear as it could be whether you are talking about too different objects.
- It appears as though the absolute homogeneity of the ReLU is an important feature of your analysis (because the of the output scaling $\sigma^L=\alpha$). Since your abstract highlights that the work relates specifically to polynomial interpolation, it should also perhaps mention that your analysis only applies to absolutely homogeneous activations.

**Strength And Weaknesses:**

**Strengths Weaknesses:**
- The data considered are all very simple. As far as I can tell, they are polynomials of order $k=1,2,3,4$. In this setting we would expect kernel methods to perform very well compared with (finite width) neural networks, even if the kernel is not a polynomials. Is there any evidence to suggest that the insights in this paper generalise to more complicated settings?
- The paper has some very nice insights concerning some very important and interesting phenomena in non-kernel regime networks. The empirical analysis seems thorough and is based on what appears to be correct mathematically motivated calculations
- However, I found it quite difficult to read. The paper feels like a set of isolated empirically established facts, rather than a cohesive and memorable story. In writing my summary, I was not able to find any single standout takeaway message that was theoretically or practically useful, beyond its immediate implication. To make matters worse, the studied situations (polynomial interpolation, MLPs, ...) are never considered in practice. Perhaps this kind of work is more suitable for a journal publication, where there is more room to establish some ground work and set the scene for the story. I note that there is lots of extra text in the appendix that is not all proof and figures.

**Summary Of The Paper:**

**POST-REBUTTAL UPDATE:**
The authors did an excellent job responding to my concerns, in particular, with the take-home message. It appears as though the take-home message was not lost on the other reviewers. In light of the author response and the other reviews, I have increased my score and decreased my confidence. I recommend this paper be accepted.

========

Often analysing wide neural networks in a kernel regime is done by choosing a finite dataset size $P$ and taking the "width" (which may actually be many "widths" in different layers) $N$ to infinity. This approach leads to a tractable analysis but can often result in a performance gap in performance between the neural network model and the kernel method model. This is usually attributed to smaller width networks being able to perform feature learning.

When the dataset size $P$ is asymptotically less than the square root of the number of parameters $\sqrt{N}$, the authors find empirically that finite-width effects are particularly strong for polynomial interpolation tasks. The transition at $P \sim \sqrt{N}$ can be delayed to higher values of $P$ through ensembling or a type of feature learning. A particularly important parameter for feature learning is the output scale $\alpha$. Feature learning may alternatively be performed by learning higher order polynomials. The authors find that ensembling according to the kernel is more effective than averaging the network predictions or features.

**Summary Of The Review:**

This paper studies a special case of the important problem of reconciling infinite width theory with the behaviour of finite width networks. The paper is informed by a rigorous empirical analysis based on mathematical truth. Unfortunately, the story it told was not memorable and the theoretical and/or practical takeaways were not clear. I think this paper could be improved by trying to synthesise the isolated conclusions into a more holistic message.

---

> ### Author Response · Authors · 2022-11-17
> **Response to Reviewer 6LDR**
>
> ### Strengths and Weaknesses:
> 1. *The data considered are all very simple. As far as I can tell, they are polynomials of order $k=1,2,3,4$. In this setting we would expect kernel methods to perform very well compared with (finite width) neural networks, even if the kernel is not a polynomials. Is there any evidence to suggest that the insights in this paper generalize to more complicated settings?*
>
> We appreciate this critique and think it is important. Polynomial regression, while analytically convenient is rather far from realistic deep learning tasks. To enhance the relevance of our study to realistic deep learning, we have performed similar ensembling experiments in Wide ResNets trained on a CIFAR image classification problems to identify the bias and variance components of test error. In the new Figure 8, we plot the test loss as function of $\alpha$ and $P$, for width (channel count) $N = 64$ CNNs. We see that ensembling gives a reduction in test error that is more prominent for lazy networks.  Like our earlier experiments, more data and smaller $\alpha$ give improvements in test error. We plot generalization error, accuracy, and the fraction of generalization error due to variance. We recover the trends observed in case of MLPs.
>
>  We will aim to do more thorough studies of the CIFAR ResNet example, adding several new phase plots and accompanying figures. For the camera-ready copy, we may aim to include these results in the main text to motivate practitioners about the variance-limited effects studied in this paper.
>
> 2. *The paper has some very nice insights concerning some very important and interesting phenomena in non-kernel regime networks. The empirical analysis seems thorough and is based on what appears to be correct mathematically motivated calculations*
>
> Thank you for this supportive comment.
>
> 3. *However, I found it quite difficult to read. The paper feels like a set of isolated empirically established facts, rather than a cohesive and memorable story. In writing my summary, I was not able to find any single standout takeaway message that was theoretically or practically useful, beyond its immediate implication. To make matters worse, the studied situations (polynomial interpolation, MLPs, ...) are never considered in practice. Perhaps this kind of work is more suitable for a journal publication, where there is more room to establish some ground work and set the scene for the story. I note that there is lots of extra text in the appendix that is not all proof and figures.*
>
> **Take home message:**
>
> We thank the reviewer for this piece of feedback. We agree that a clearly stated message could reinforce the overall story of the work and provide coherence to the set of results we enumerate in the contributions list and empirical result list. To be explicit about our take-home message, we added the following paragraph below the contributions list on page 2
>
> > Overall, our results indicate that the onset of finite-width corrections to generalization in neural networks become relevant when the scale of the variance of kernel fluctuations becomes comparable to the bias component of the generalization error in the bias-variance decomposition. The variance contribution to generalization error can be reduced both through ensemble averaging and through feature learning, which we show promotes higher alignment between the final kernel and the task. We construct a model of noisy random features which reproduces the essential aspects of our observations.
>
> Similarly, on page 3 below our list of empirical findings, we write
> > These empirical findings support our hypothesis that finite width introduces fluctuations in kernels over initializations, which ultimately leads to variance in the learned predictor and higher generalization error.
>
> **Practical Networks:**
>
> As we mentioned above, we have incorporate more realistic networks and learning tasks (CNNs trained on CIFAR).
>
> In addition, we have been trying to study polynomial regression tasks where the targets are mixtures of different degree spherical harmonics. Studying such mixtures is useful since any function on the sphere can be broken up into such a decomposition. We find that, at least in the lazy regime, the bias and variance for the mixture task can be computed as a simple linear combination of the biases and variances of the error for separate target functions of different degrees. In Figure 16 we compare the bias and variance of networks trained on a mixed $k=1, k=3$ target function with the sum of the respective biases and variances of networks trained on $k=1$ and $k=3$ modes separately. We find that in the lazy regime the mixed mode learning curves are recovered directly as a linear combination of the pure mode curves.
>
> We have additionally included new plots sweeping over depth and input dimension in appendix figures 13, 14. We hope this this increase in experimental scope will make for a more compelling presentation.

---

> > ### Author Response · Authors · 2022-11-17
> > **Response to Reviewer 6LDR Part 2**
> >
> > **Content in Appendix**
> >
> > While there is additional detail about experiments and learning curve computations for the toy model, we tried to include most of the key content and takeaways in the main text. While more space would be helpful, we believe our main message can be communicated in 9 pages with additional experimental and theoretical detail for the interested reader relagated to the appendix. If the reviewer has suggestions on which elements of the appendix to reduce, or to move to the main text we would be happy to incorporate them.
> >
> >
> > ### Questions:
> > 1. *Abstract. "However, at a critical sample size $P^∗$, the finite-width network generalization begins to worsen compared to the infinite width performance." Is it that the generalization necessarily worsens, or that the generalization bounds that have been proven are worse than the generalization bounds for the corresponding infinite width model?*
> >
> > We rewrote this sentence to emphasize that this is am empirical finding that we subsequently aimed to explain theoretically. This sentence now reads:
> >
> > > However, at a critical sample size $P^*$, we empirically find the finite-width network generalization begins to worsen compared to the infinite width performance.
> >
> > 2. *"we demonstrate empirically that all networks have the same generalization error as kernel regression solutions with their final eNTKs. A". It is not obvious to me why networks would \emph{not} have the same generalization error as kernel regression using final eNTKs. Can you maybe add a sentence explaining why this is the case? Perhaps in general one would expect the kernel regression to be closer to the GPNN kernel (in the sense of Neal, Matthews, Lee, etc.) rather than the NTK (Jacot etc.)?*
> >
> > This is a good question and one that several other papers have investigated (Fort 2020 https://arxiv.org/abs/2010.15110, Atanasov et al 2021 https://arxiv.org/abs/2111.00034, Vyas et al 2022 https://arxiv.org/abs/2206.10012). In general, the learned function contains contributions from the instantaneous NTKs at every point in the training. Concretely, following Atanasov et al, we have the following formula for the final network predictor $f(x)$
> >
> > $$f(x) = \int_0^\infty dt \mathbf k(x,t) \cdot \exp\left( - \int_0^t ds \mathbf K(s) \right) \mathbf y$$
> >
> > where $[\mathbf k(x,t)]_\mu = K(x, x_\mu, t)$ and $[\mathbf K(s)]_\mu ,_\nu= K(x_\mu, x_\nu, s)$  and $[\mathbf y]_\mu = y_\mu $.
> >
> > The problem is that the function $f$ cannot always be written as a linear combination of the final NTK on training data ie $f = \sum_{\mu} \alpha_\mu K_f(x,x_\mu)$ where $K_f$ is the final NTK since there are contributions from earlier kernels $\mathbf k(x,t)$ for $t < \infty$. However, as Atanasov et al, Vyas et al and other show, the final predictions of the network are often well modeled by regression with the final NTK.
> >
> > We have added a comment about this fact in Appendix C.4 to add clarity about why we focus on making sure the final kernel agrees with the trained network.
> >
> > 3. *Around equation 2. "There we also show that the O(1/N) term is strictly positive for sufficiently large N. Thus, for lazy networks, finite width effects lead to strictly worse generalization error." Wouldn't it be more appropriate to say that "for lazy networks \emph{of sufficiently large $N$}, finite widht effects lead to strictly worse generalization error." Since the error term is strictly positive only for sufficiently wide networks, less wide networks might have better generalisation error.*
> >
> > Yes, thank you for helping us clarify our statement here. We meant to say exactly that for sufficiently large $N$, smaller width improves generalization. We have made this change.
> >
> >
> > ### Minor:
> > * *Last sentence of conclusion. "and number of ensembles of a network can be tuned to achieve optimal performance under a fixed compute and data budget." Should this say something more like "number of networks in an ensemble"?*
> >
> > Indeed, the correct wording should be "number of networks in an ensemble". We have fixed this error in language.
> >
> > * *Somewhere at the beginning, you introduce the terminology "final empirical NTK" with a symbol like eNTK_f. Instead of repeating "final empirical NTK", you could just write the symbol eNTK_f. Alternatively, repeat the terminology. At the moment, you use both (eg. the next dot point after you introduce the terminology). If you repeat both, it is not as clear as it could be whether you are talking about too different objects.*
> >
> > Thank you for noticing this. We have taken care to update the text and write eNTK_f in all places instead of "final kernel" to avoid readers confusing these for two different objects

---

> > > ### Author Response · Authors · 2022-11-17
> > > **Response to Reviewer 6LDR Part 3**
> > >
> > > * *It appears as though the absolute homogeneity  of the ReLU is an important feature of your analysis (because the of the output scaling ). Since your abstract highlights that the work relates specifically to polynomial interpolation, it should also perhaps mention that your analysis only applies to absolutely homogeneous activations.*
> > >
> > > We thank the reviewer for bringing this to our attention. While our empirical experiments focused on ReLU networks, and our current implementation of output rescaling through weight rescaling, we show that this is not strictly necessary to alter the level of feature learning. We added Appendix C.2 which describes an alternative parameterization which works for nonhomogenous activations as well as homogenous activations. We refer to this section in the main text Section 2 under equation (1). Indeed, we use such an alternative parameterization in our training of the ResNets in Figure 8.

---

> > > > ### Comment · Reviewer_6LDR · 2022-11-17
> > > > **Response to authors**
> > > >
> > > > Thanks for your detailed response to my feedback. I have updated my review accordingly --- see above.

---

### Official Review · Reviewer_PWNi · 2022-10-31

**Confidence:** 4
**Clarity, Quality, Novelty And Reproducibility:** See above.
**Correctness:** 4
**Technical Novelty And Significance:** 4
**Empirical Novelty And Significance:** Not applicable
**Recommendation:** 8

**Strength And Weaknesses:**

The paper tackles an important topic: given the ubiquity of papers studying
neural nets in the infinite-width limit, it is important to understand the
limitations of this regime in practice. While previous work (which the authors discuss)
had already identified the variance-limited regime, or discussed the role of ensembling,
here the authors put everything together in a single case-study.

The paper is also well-written: it clearly states the setup and the results, and
it connects nicely with the previous literature.

**Summary Of The Paper:**

*Please note that this is an emergency review, so I was not able to go as much into details of the paper as I would have liked to*

A lot of recent work on deep learning theory has focused on understanding the
dynamics and generalization of neural networks with infinitely wide hidden
layer. This is of course an idealisation, and recent work has sought to
understand when finite-width effects become relevant as the size of the training
set increases.

This work performs a careful study of two-layer ReLU networks trained on a
simple polynomial regression task to investigate precisely when finite-size
effects become important for the performance of the network. The authors vary
three key parameters: the width of the network, $N$, the size of the training
set $P$, and the *scale* of the network $\alpha$, which was introduced by Chizat
et al. (2019) as a mean to interpolate between the "lazy regime" (large
$\alpha$) and the "feature learning regime" (small $\alpha$).

The authors find through experiments that for large data sets, the
infinite-width limit always performs best. Feature learning does improve over
the infinite-width limit at intermediate data set size, while lazy learning at
intermediate sizes does not. They relate these behaviours to the fluctuations due to
various sources of noise in the system (for example in the initialisation of the weights),
and show how ensembling can mitigate these fluctuations. The authors propose various scaling laws to explain
their findings, which they obtain via a toy model which qualitatively reproduces
the effects (cf. Fig 6).


**Summary Of The Review:**

To summarise, this paper provides a careful investigation of a problem which is
of interest to the theory of neural networks community, and should therefore be
accepted at ICLR.

---

> ### Author Response · Authors · 2022-11-17
> **Response to Reviewer PWNi**
>
> We thank the reviewer for their time and close reading, especially as they were an emergency review. We appreciate all of the supportive comments about our study of the onset of finite size effects in neural network learning curves. We are pleased to hear that the reviewer found our work relevant and our writing clear. In the latest draft we have expanded the scope of our experiments and have added additional information in the appendix about deriving learning curves using our toy model.

---

> > ### Comment · Reviewer_PWNi · 2022-11-17
> > **Acknowledging the other reviews and the replies**
> >
> > This is just to acknowledge that I read the other reviews as well as the authors' replies. I appreciate the effort of the authors to address some of the concerns of the other reviewers, in particular the experiments on the Wide ResNet. I would even encourage the authors to put these results more prominently in the main text. In any case, I am happy to maintain my score and to recommend acceptance.

---

### Decision · Program_Chairs · 2023-01-20

**Decision:**

Accept: poster

**Justification For Why Not Higher Score:**

Still limitation to data / model: In the rebuttal authors showed WideResNet on CIFAR-10 as well as more experiments on input dimension and depth showing consistent results as the main message of the paper. While this shows the main idea is not limited to the simple set up, exploration to practical cases still is not sufficient.

**Justification For Why Not Lower Score:**

Reviewer `PWNi`: "this paper provides a careful investigation of a problem which is of interest to the theory of neural networks community, and should therefore be accepted at ICLR."

Still an interesting and  comprehensive study of neural networks in the lazy and rich regimes to infinite width NTK at different dataset size scales.

**Metareview: Summary, Strengths And Weaknesses:**

The paper studies the generalization property of finite width networks to their infinite width counterparts. Authors empirically analyze the effect of changing the dataset size (P) for fixed networks. Authors argue finite width effects depend whether the network is in a lazy or rich regime.


Strength
- With recent interest in infinite width networks, it is essential to understand and characterize discrepancy between finite vs infinite networks. The paper makes a  number of interesting observations in that regard supported by detailed experiments and a toy model.
- Clearly written and figure are clear
- Good coverage of literature and well placed
- Provides complete overview of lazy and feature learning regime.
- Experiments are done in high quality

Weakness
- The effect of depth and task dependence (input dimension, degree of polynomial, general function class)  is not captured in the study which is quite important for deep learning.
- Structure of the data is limited: degree k polynomial on the sphere with uniformly distributed inputs
- Architecture is limited to MLP and Conv may be different

Authors promised ("We plan to make the results of our experiments publicly accessible, alongside the code to generate them.") experiments to be publicly available. AC encourages the authors to deliver on the promise to help reproducibility of the results.




**Note From Pc:**

if the above contains the word "oral" or "spotlight" please see: "oral" presentation means -> notable-top-5% and "spotlight" means -> notable-top-25%. As stated in our emails, we are disassociating presentation type from AC recommendations